# Controllable Logical Hypothesis Generation for Abductive Reasoning in Knowledge Graphs

**Yisen Gao, Jiaxin Bai, Tianshi Zheng& Yangqiu Song**
Department of Computer Science and Engineering
The Hong Kong University of Science and Technology
Hong Kong, China
`{ygaodi,jbai,tzhengad,yqsong}@cse.ust.hk`

**Ziwei Zhang , Qingyun Sun & Jianxin Li**
Department of Computer Science and Engineering
Beihang University
Beijing, China
`{zwzhang,sunqy,lijx}@buaa.edu.cn`

**Xingcheng Fu**
Key Lab of Education Blockchain and Intelligent Technology
Guangxi Normal University
Guangxi, China
`fuxc@gxnu.edu.cn`

## Abstract

Abductive reasoning in knowledge graphs aims to generate plausible logical hypotheses from observed entities, with broad applications in areas such as clinical diagnosis and scientific discovery. However, due to a lack of controllability, a single observation may yield numerous plausible but redundant or irrelevant hypotheses on large-scale knowledge graphs. To address this limitation, we introduce the task of controllable hypothesis generation to improve the practical utility of abductive reasoning. This task faces two key challenges when controlling for generating long and complex logical hypotheses: hypothesis space collapse and hypothesis reward oversensitivity. To address these challenges, we propose **CtrlHGen**, a **C**ontrollable **l**ogcial **H**ypothesis **Gen**eration framework for abductive reasoning over knowledge graphs, trained in a two-stage paradigm including supervised learning and subsequent reinforcement learning. To mitigate hypothesis space collapse, we design a dataset augmentation strategy based on sub-logical decomposition, enabling the model to learn complex logical structures by leveraging semantic patterns in simpler components. To address hypothesis reward oversensitivity, we incorporate smoothed semantic rewards including Dice and Overlap scores, and introduce a condition-adherence reward to guide the generation toward user-specified control constraints. Extensive experiments on three benchmark datasets demonstrate that our model not only better adheres to control conditions but also achieves superior semantic similarity performance compared to baselines. Our code is available at https://github.com/HKUST-KnowComp/CtrlHGen.

## 1 Introduction

Abduction is widely recognized as one of the three major types of reasoning in philosophy (Douven, 2011). Specifically, abductive reasoning (Douven, 2011) is a form of logical inference that seeks the best or most plausible hypothesis to explain an observed phenomenon and it plays a vital role across various fields (Paul, 1993). For example, it serves as a critical tool for hypothesizing causal links between symptoms and underlying pathologies in clinical diagnosis (Pukancová & Homola, 2015; Martini, 2023). Similarly, abductive methods localize system faults by interpreting anomalous signal patterns in anomaly detection (Ramkumar et al., 2024; Ganesan et al., 2019). Its power also extends to scientific discovery (Engelschalt et al., 2023; Wackerly, 2021; Duede & Evans, 2021; Upmeier zu Belzen et al., 2021), including the deduction of unknown celestial bodies from gravitational perturbations in orbital trajectories (Smart, 1946).

On the other hand, effective abductive reasoning requires high-quality, interconnected information. While large language models perform well in common-sense settings (Patil & Jadon, 2025), they

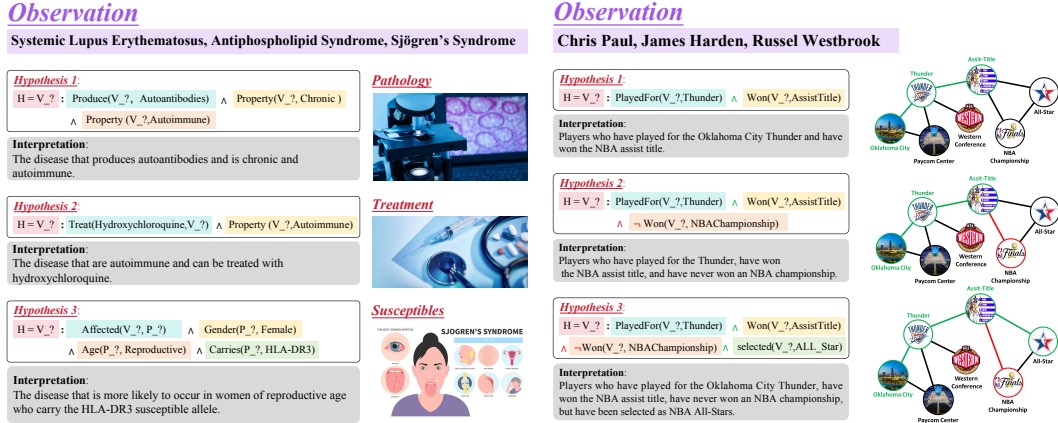

Figure 1: Examples of Controllability in Abductive Reasoning

often struggle in domains such as healthcare, business, or other scenarios involving sensitive data and strict privacy constraints. Knowledge graphs, whether general-purpose or domain-specific, provide a structured foundation that supports more reliable abductive reasoning. In knowledge graphs, abductive reasoning aims to generate complex logical hypotheses that explain observed entities, leveraging domain knowledge to improve inference precision and reliability. AbductiveKGR (Bai et al., 2024b) was the first to introduce this task, formulating it as logical query generation over structural knowledge and training models through a supervised–reinforcement learning framework.

However, knowledge graphs often contain millions of facts, which can lead to generate numerous plausible but irrelevant hypotheses from a single observation. For instance, even the relatively small DBpedia50 dataset(with only 24,624 entities and 351 relations), produces an average of 50 reasonable hypotheses per observation. In larger graphs, this number grows dramatically, underscoring the need to filter hypotheses according to user intent or interests for effective abductive reasoning. To address this challenge, we introduce controlling mechanisms into the hypothesis generation process, focusing on two critical aspects:

*Controlling semantic content enables aspect-specific reasoning.* We prioritize semantic control to narrow vast hypothesis spaces to relevant aspects, essential for specialized fields where aspect-specific insights drive decision-making. As shown in Fig. 1a, we want to explain the observation involving three diseases:{Systemic Lupus Erythematosus, Antiphospholipid Syndrome, and Sjögren's Syndrome}. Directing attention to specific aspects—such as pathology, treatment, or affected populations—yields hypotheses that are precisely aligned with each aspect. From the pathology aspect, these diseases produce autoantibodies and are both chronic and autoimmune. From the treatment aspect, these autoimmune diseases can be treated with hydroxychloroquine. Finally, from the susceptibility aspect, these diseases are more likely to occur in women of reproductive age who carry the HLA-DR3 susceptible allele. Although these hypotheses are all plausible, their usefulness varies when people seek explanations for different scenarios.

*Controlling structural complexity adjusts the level of granularity.* We focus on complexity control to address varying information needs across different reasoning scenarios and align with users' cognitive preferences for adjustable information density. In Fig. 1b, for an observation composed of three NBA players, increasing the complexity of the hypothesis structure enables the model to capture richer shared experiences or achievements among them. By adjusting the structural complexity, users can flexibly decide how much information they want to include in the generated hypotheses. Unfortunately, prior work (Bai et al., 2024b) has largely overlooked controllable generation, resulting in hypotheses that are redundant or lack meaningful relevance.

Motivated by these, we introduce the task of controllable abductive reasoning, aiming at controllable generation of hypothesis, which leads to better leverage the practical value of abductive reasoning in knowledge graphs. However, when implementing semantic and structural controls on complex long logical hypotheses, we face two critical challenges: (i) Hypothesis Space Collapse: As illustrated in Fig. 2a, the number of plausible hypotheses drops sharply as their length increases. This sharp decline severely limits our ability to apply structural complexity control, as the model needs to ensure a

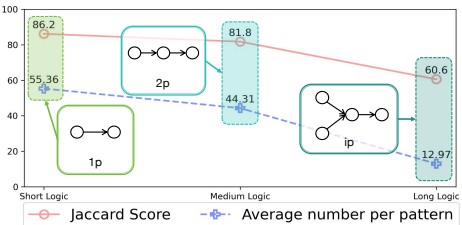
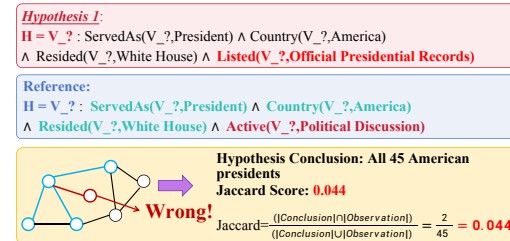

(a) Hypothesis Space Collapse        (b) Hypothesis Reward Oversensitivity

Figure 2: (a) Hypothesis quality (measured in Jaccard) and space size across three logic lengths: short (one predicate), medium (two predicates), and long (three predicates). Valid candidates represent average reference hypotheses per observation. Note the dramatic collapse of hypothesis space as complexity increases. (b) Hypothesis oversensitivity example: Minor errors cause significant Jaccard score drops, creating tension between control adherence and semantic accuracy.

strong understanding of complex logic in order to make correct candidate hypotheses. (ii) Hypothesis Reward Oversensitivity: The previous approach (Bai et al., 2024b) utilized the Jaccard score as a reward mechanism to enhance the model's understanding of query semantics. However, as illustrated in Fig. 2b, during the model's exploration process, even a minor misstep may lead to a sharp drop in the Jaccard score, severely disrupting training stability and guiding the model toward incorrect directions.

To tackle these challenges, we propose a **Cont**rollable **l**ogcial **H**ypothesis **Gen**eration method (**CtrlHGen**) for abductive reasoning in knowledge graphs. To address the problem of hypothesis space collapse, we introduce a dataset augmentation strategy based on sub-logical decomposition. By leveraging the semantic similarity of simpler sub-logics derived from the decomposition of complex hypotheses, this approach enables the model to understand long logical structures, which are composed of these smaller components. The hypothesis generator is then trained using a combination of supervised fine-tuning and reinforcement learning. To address the problem of hypothesis reward oversensitivity, we refine the semantic reward function by incorporating Dice and Overlap coefficient to smooth out minor discrepancies between the hypothesis and the target. Additionally, we introduce a condition-adherence reward to encourage the generation of hypotheses that adhere to the control constraints during exploration. Our main contributions are as follows:

- We are the first to introduce the task of controllable abductive reasoning, enabling abductive reasoning in knowledge graphs to better satisfy practical needs by controlling semantic content and structural complexity.
- We propose an observation-hypothesis pair augmentation strategy via sub-logical decomposition to address the challenge of hypothesis space collapse when generating complex logical structures, significantly enhancing the quality of controllable hypotheses.
- To mitigate hypothesis reward oversensitivity, we refine the semantic reward function by incorporating Dice and Overlap coefficients to accommodate minor discrepancies between hypotheses and targets, while introducing a condition-adherence reward to ensure better compliance with control constraints, leading to more stable and accurate learning.
- Extensive experiments on three datasets demonstrate that our model not only adheres more effectively to control signals but also achieves superior semantic similarity performance compared to the baseline across multiple evaluation metrics.

## 2   RELATED WORK

**Knowledge Graph Reasoning**. Deductive reasoning focuses on answering complex logical queries by improving query and answer embeddings (Zhang et al., 2021; Ren et al., 2020; Bai et al., 2022; 2023a;b; 2024a). Inductive reasoning, often framed as rule mining, ranges from efficient symbolic methods like AMIE (Galárraga et al., 2013) to embedding-based approaches such as RuLES (Ho et al., 2018) and RLogic (Cheng et al., 2022), though traditional search-based techniques face

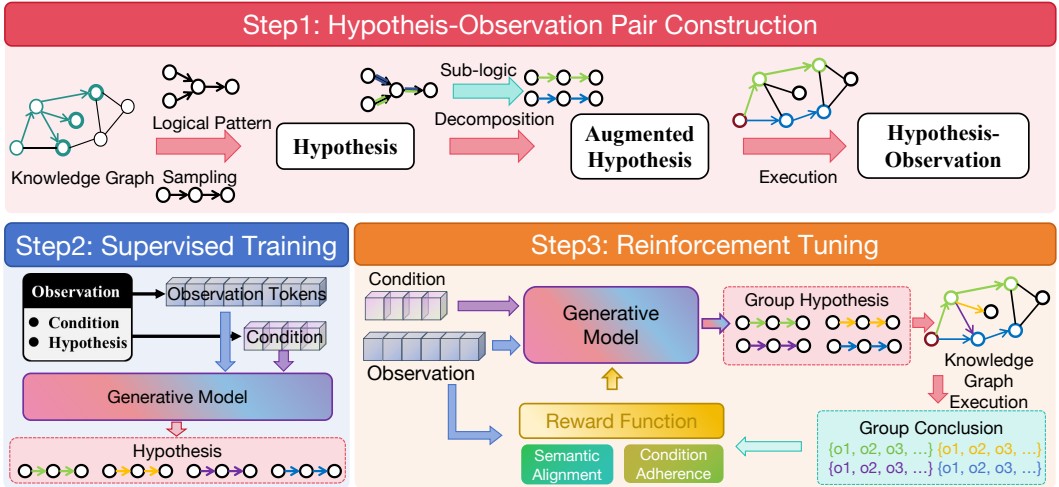

Figure 3: An overview of our controllable abductive reasoning framework. The process consists of three main steps: (1) Hypothesis-Observation pair construction through sub-logic decomposition to expand the hypothesis space, (2) Supervised training of the generative model using augmented hypotheses, and (3) Reinforcement tuning with dual rewards for semantic alignment and condition adherence to balance hypothesis accuracy with control signal compliance.

scalability challenges. Abductive reasoning was introduced by AbductiveKGR (Bai et al., 2024b) using Transformer-based hypothesis generation, with follow-up work (Bai et al., 2025) highlighting its future potential.

**Abductive Reasoning**. In natural language inference, $\alpha$-NLI (Bhagavatula et al., 2020) introduced abductive reasoning to commonsense reasoning, where plausible explanations are inferred from observations. Subsequent works proposed various techniques to enhance this capability (Qin et al., 2021; Kadiķis et al., 2022; Chan et al., 2023), including extensions to uncommon scenarios focusing on rare but logical explanations (Zhao et al., 2024). Unlike real-world data in commonsense reasoning, benchmarks like ProofWriter (Tafjord et al., 2021) evaluate formal abductive reasoning within semi-structured texts with explicit logical relationships. Recent studies have explored LLMs in more challenging open-world reasoning contexts (Zhong et al., 2023; Del & Fishel, 2023; Thagard, 2024) and abstract reasoning tasks (Liu et al., 2024b; Zheng et al., 2025).

Meanwhile, in the neuro-symbolic domain, Abductive Learning (ABL) (Zhou, 2019) attempts to integrate machine learning and logical reasoning in a balanced and mutually supportive manner. Recent research in this area, exemplified by systems such as ARLC (Camposampiero et al., 2024) and ABL-Refl (Hu et al., 2025), focuses on enhancing this integration by introducing novel techniques to improve context-awareness, error correction, generalization, and overall reasoning accuracy and efficiency.

## 3 METHOD

In this section, we elaborate the proposed CtrlHGen, a controllable hypothesis generation method for abductive reasoning in knowledge graphs. The framework of CtrlHGen is shown in Fig. 3.

### 3.1 PROBLEM DEFINITION

We define a knowledge graph as $G = (V, R)$, where $V$ is the set of entities and $R$ is the set of binary relations. A triple $(u, r, v)$ exists in $G$ if $r(u, v) = \text{true}$. Following the open-world assumption (Drummond & Shearer, 2006), only the observed graph $G$ is available during training, with missing triples treated as unknown rather than false. The full graph $\bar{G}$ remains hidden, and $G \subseteq \bar{G}$.

The core concepts of abductive reasoning consist of observation and hypothesis. Here, an observation $O$ in knowledge graph $G$ is defined as a set of entities $O = \{o_1, o_2, \ldots, o_n\}$, where $o_i \in V, \forall i \in \{1, \ldots, n\}$ . A logical hypothesis $H$ is defined as a query in the form of first-order logic on a knowledge graph $G$, including existential quantifiers($\exists$), And($\wedge$), Or($\vee$), Not($\neg$). The hypothesis can also be written in disjunctive normal form:

$$H(V_?) = \exists V_1, \ldots, V_k : e_1 \vee \cdots \vee e_n,$$
$$e_i = r_{i1} \wedge \cdots \wedge r_{im_i}, \tag{1}$$

where $\{V_1, \ldots, V_k\}$ denotes the subset of $V$. Each $r_{ij}$ is defined as either $r_{ij} = r(u, v)$ or $r_{ij} = \neg r(u, v)$, where $u$ and $v$ are either fixed entities from the set $\{V_1, \ldots, V_k\}$, or variable vertices $V_?$, which could be any entity on the graph $G$.

The conclusion of the hypothesis $[H]_G$ on a graph $G$ is the set of the variable entities $V_?$ for which $H$ holds true on $G$. Specifically, it can be formulated as:

$$[H]_G = \{V_? \in G | H(V_?) = \text{true}\}. \tag{2}$$

**Definition 3.1** (Controllable Abductive Reasoning in Knowledge Graph). Given a knowledge graph $G$, an observation $O$, and a control condition $C$, the goal of *controllable abductive reasoning* is to find a hypothesis $H$ satisfying:

1. The hypothesis $H$ is the most plausible explanation for the observation $O$ . In other words, the conclusion $[H]_G$ closely matches the observation $O$.

2. $H$ satisfies the constraints specified by the control condition $C$.

## 3.2 Observation-Hypothesis Pairs Construction

**Sampling**. We randomly sample observation-hypothesis pairs from the knowledge graph by constructing hypotheses based on predefined logical patterns. Each logical pattern is assigned an equal number of hypotheses to ensure diversity, and the conclusion of hypotheses on the graph are taken as the corresponding observations. Finally, both hypotheses and observations are converted into input sequences suitable for the generative model.

**Augmentation by sub-logic decomposition**. To address the challenge of hypothesis space collapse in complex logical patterns, we propose a dataset augmentation method based on sub-logic decomposition. Specifically, given a hypothesis–observation pair $(H, O)$ under a complex logical pattern $P$, we recursively decompose the hypothesis into sub-hypotheses $H_{\text{sub}}$ according to identifiable sub-logical patterns $P_{\text{sub}}$. Corresponding sub-observations $O_{\text{sub}}$ are then derived by executing these sub-hypotheses on the knowledge graph $G$. This process effectively generates additional hypothesis–observation pairs and can be formally described as:

$$\{(H_{\text{sub}}^i, O_{\text{sub}}^i)\}_{i=1}^n = \left\{ (f(P_{\text{sub}}^i, H), [f(P_{\text{sub}}^i, H)]_G) \mid P_{\text{sub}}^i \subseteq P \right\}, \tag{3}$$

where $f(P_{\text{sub}}^i, H)$ denotes the sub-hypothesis generated based on the sub-pattern $P_{\text{sub}}^i$ and the origin hypothesis $H$, and $[f(P_{\text{sub}}^i, H)]_G$ computes the corresponding sub-observation by querying the knowledge graph to get the conclusion of the sub-hypothesis.

Because each sub-hypothesis is a subset of the original, they are closely related both structurally and semantically. This strong alignment enables the model to progressively learn complex logical patterns by building on simpler, related sub-patterns. We have reported more details in Appendix A.

## 3.3 Supervised Training of Controllable Hypothesis Generation

To enable controllable generation of logical hypotheses, we train a conditional generative model to generate hypothesis sequences guided by a given observation and control condition. Specifically, given an observation sequence $O = \{o_1, \ldots, o_m\}$, a target hypothesis sequence $H = \{h_1, \ldots, h_n\}$, and a control condition $C$, the generative model is optimized using an autoregressive loss:

$$\mathcal{L}_{\text{AR}} = -\log p_\theta(H \mid O, C) = -\sum_{i=1}^n \log p_\theta(h_i \mid h_1, \ldots, h_{i-1}, O, C), \tag{4}$$

where $\theta$ denotes the generative model, which we implement using a standard Transformer-based decoder-only architecture.

The training process consists of two stages. In the first stage, the model is trained under an unconditional setting, where the input only consists of observation tokens. This allows the model to acquire a general capability for hypothesis generation. In the second stage, the model is fine-tuned under different control conditions respectively. The input is formed by concatenating observation tokens with control condition tokens, guiding the model to generate hypotheses that satisfy the constraints.

The control conditions $C$ are designed from two different perspectives to guide hypothesis generation:

- **Semantic Focus**: We randomly sample a specific entity or relation from the target hypothesis as a control condition. This guides the model to generate hypotheses grounded in a specific semantic region of the knowledge graph. The control condition is directly represented by the token of the selected entity or relation. Formally, $C \in \{T_e\}$ or $C \in \{T_r\}$. $T_e$ and $T_r$ represents the token set of entity and relation respectively.

- **Structural Constraint**: We apply constraints based on the logic structure of the hypothesis. Specifically, we implement three types of structural control: (1) strictly enforcing a predefined logical pattern, where each logical pattern is represented in Lisp-like language with operator tokens following previous work in KG reasoning (Bai et al.; 2024b). (2) constraining the number of entities involved, encoded using a special token [ne] that indicates hypotheses with exactly $n$ entities. Formally, $C \in \{[ne]\}$, where $n$ is an Integer. (3) constraining the number of relations used in the generated hypothesis, encoded using a token [nr], where [nr] denotes hypotheses containing exactly $n$ relations. Formally, $C \in \{[nr]\}$, where $n$ is an Integer.

## 3.4 REINFORCEMENT LEARNING

To improve the generalization ability on unseen knowledge graphs and better adhere to the specified control conditions, we further fine-tune the generative model using reinforcement learning. The reward function is constructed from two perspectives: semantic alignment and condition adherence.

**Semantic Alignment**: This reward assesses the semantic consistency between the generated hypothesis conclusion $[H]_G$ and the corresponding observation $O$. We adopt the Jaccard similarity coefficient as the primary reward due to its strict evaluation of set-level agreement. However, the high sensitivity of hypotheses can lead to sharp reward fluctuations in response to minor errors. To mitigate this, we integrate two supplementary metrics: the Dice similarity coefficient and the Overlap similarity coefficient, which provide smoother gradients and greater tolerance to slight mismatches. The final semantic reward $R_{\text{sem}}$ is computed as a weighted combination of these metrics, defined as:

$$
\begin{aligned}
R_{\text{sem}}([H]_G, O) &= \lambda_1 \cdot \text{Jaccard}([H]_G, O) + \lambda_2 \cdot \text{Dice}([H]_G, O) + \lambda_3 \cdot \text{Overlap}([H]_G, O) \\
&= \lambda_1 \cdot \frac{|[H]_G \cap O|}{|[H]_G \cup O|} + \lambda_2 \cdot \frac{2|[H]_G \cap O|}{|[H]_G| + |O|} + \lambda_3 \cdot \frac{|[H]_G \cap O|}{\min(|[H]_G|, |O|)},
\end{aligned}
\tag{5}
$$

where $\lambda_1$, $\lambda_2$, and $\lambda_3$ are hyperparameters. $G$ denotes the observable knowledge graph during training, which serves as a reliable and leakage-free proxy for evaluating abductive reasoning quality.

**Condition Adherence**: This reward encourages the model to generate hypotheses that satisfy the given control condition $C$. We formulate it as a binary-valued function: if the generated hypothesis $H$ satisfies the condition $C$, the reward is 1; otherwise, it is 0. The final adherence performance is evaluated by computing the proportion of generated hypotheses that meet the condition. Formally, the reward function is defined as:

$$
R_{\text{cond}}(H, C) = \begin{cases} 1, & \text{if } H \text{ satisfies } C, \\ 0, & \text{otherwise.} \end{cases}
\tag{6}
$$

Jointly capturing condition adherence and semantic alignment, the overall reward function $\hat{R}$ is formulated as:

$$
\hat{R}(H, O, C, G) = \alpha \cdot R_{\text{sem}}([H]_G, O) + (1 - \alpha) \cdot R_{\text{cond}}(H, C),
\tag{7}
$$

where $\alpha \in [0, 1]$ is a hyperparameter that balances the contributions of semantic alignment and condition adherence.

Since abductive reasoning often involves generating multiple plausible hypotheses rather than a single answer, it is important to ensure overall hypothesis quality. To achieve this, we use Group Relative

Policy Optimization (GRPO) (Shao et al., 2024), which promotes consistent improvement across a set of sampled hypotheses per observation, instead of optimizing individual outputs. Specifically, GRPO updates the model $\pi_\theta$ by maximizing the expected reward over a group of hypotheses $\hat{H} = H_1, \ldots, H_k$ sampled from the same observation $O$ and control condition $C$. The objective is:

$$\mathcal{J}(\theta) = \mathbb{E}_{O,\{H_i\} \sim \pi_{\theta_{\text{old}}}(H|O,C)} \left[ \frac{1}{k} \sum_{i=1}^{k} \frac{1}{|H_i|} \sum_{t=1}^{|H_i|} \left\{ \frac{\pi_\theta(h_{i,t}|O,C,h_{i,<t})}{\pi_{\theta_{\text{old}}}(h_{i,t}|O,C,h_{i,<t})} \hat{R}'_i - \beta \mathrm{D}_{KL} \left[ \pi_\theta || \pi_{\text{ref}} \right] \right\} \right],$$
(8)

where $k$ is the number of sampled hypotheses per observation. The normalized reward $\hat{R}'_i$ is obtained by applying intra-group normalization over $\{\hat{R}_1, \ldots, \hat{R}_k\}$. A KL term constrains the policy $\pi_\theta$ from drifting too far from the reference model $\pi_{\text{ref}}$, with $\beta$ controlling its strength. Gradient clipping is also used to stabilize training.

## 4 EXPERIMENT

### 4.1 EXPERIMENT SETTINGS

**Dataset**. We conduct experiments on three widely used knowledge graph datasets: DBpedia50 (Auer et al., 2007), WN18RR (Bordes et al., 2013), and FB15k-237 (Toutanova & Chen, 2015). Following (Bai et al., 2024b), each dataset is split into training, validation, and test sets with an 8:1:1 ratio. Under the open-world assumption, we incrementally build $G_{\text{train}}$, $G_{\text{valid}}$, and $G_{\text{test}}$, where each graph includes all previous edges.

**Observation-Hypothesis Pair**. Following prior KG reasoning work (Ren et al., 2020), we adopt the 13 predefined logical patterns in Fig. 4 for hypothesis sampling. Each observation contains no more than 32 entities. To evaluate generalization, the validation and test sets include entities not seen during training, with the test set covering more unseen entities. For sub-logic decomposition, we chose five complex logical patterns (up, 3in, pni, pin, inp) to break down.

**Evaluation Metrics**. The quality of generated hypotheses is evaluated in terms of semantic similarity and condition adherence. For semantic similarity, we use Jaccard, Dice and Overlap score, with $G_{\text{test}}$ used to compute $[H]_{G_{\text{test}}}$ during testing. For condition adherence, we regard it as a binary classification problem and calculate Accuracy. In addition, Smatch score (Cai & Knight, 2013) is also used to quantify the structural similarity corresponding to the generated hypothesis $H$ and the reference hypothesis $H_{\text{ref}}$. It can measure how similar the nodes, edges and their labels are by representing the hypothesis

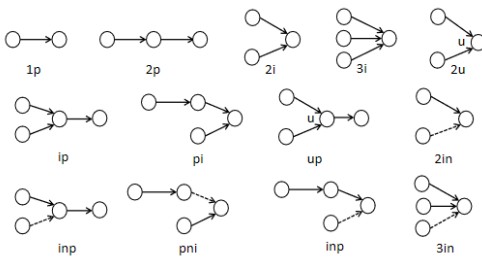

Figure 4: Thirteen predefined logical types.

as a graph. It is an evaluation metric for Abstract Meaning Representation (AMR) graphs, which are directed acyclic graphs with two node types (variable and concept) and three edge types (instance, attribute, and relation). Given a predicted graph $G_p$ and a gold graph $G_g$, Smatch($G_p, G_g$) is computed by finding an approximately optimal mapping between the variable nodes of the two graphs and matching their edges. Following the settings of Bai et al. (2024b), we transform the hypothesis graph $G(H)$ into an AMR graph $GA(H)$ by adding virtual nodes and instance edges, and then calculate Smatch. In short, Smatch is used to measure the degree of similarity between the generated hypothesis and the ground truth in the test set. It should be noted that Smatch is only a reference metric, as the generated hypotheses do not need to be the same as the reference hypotheses.

**Implementation Details**. We adopt a 12 layers decoder-only Transformer architecture (Radford et al., 2019; Vaswani et al., 2017) for the hypothesis generation model and use the AdamW optimizer. All experiments are conducted on 4 Nvidia A6000 48GB GPUs. Additional hyperparameter settings and other experiment details are reported in Appendix B.

Table 1: The results of controllable abductive reasoning under different conditions. (Result: average score $\pm$ standard deviation. **Bold**: best; Underline: runner-up. —: cannot be evaluated.)

| Dataset | Condition | Semantic Similarity | | | Condition Adherence | |
|---|---|---|---|---|---|---|
| | | Jaccard | Dice | Overlap | Accuracy | Smatch |
| FB15k-237 | uncondition | $61.4_{\pm0.33}$ | $69.3_{\pm0.31}$ | $82.3_{\pm0.33}$ | — | $61.4_{\pm0.21}$ |
| | pattern | $\mathbf{65.5}_{\pm0.33}$ | $\mathbf{73.0}_{\pm0.30}$ | $\mathbf{83.9}_{\pm0.27}$ | $98.9_{\pm0.10}$ | $82.3_{\pm0.10}$ |
| | relation-number | $65.1_{\pm0.33}$ | $72.7_{\pm0.31}$ | $83.5_{\pm0.29}$ | $99.4_{\pm0.14}$ | $\mathbf{82.4}_{\pm0.20}$ |
| | entity-number | $63.1_{\pm0.33}$ | $71.5_{\pm0.30}$ | $82.7_{\pm0.28}$ | $86.3_{\pm0.02}$ | $65.7_{\pm0.10}$ |
| | specific-entity | $64.3_{\pm0.35}$ | $71.1_{\pm0.33}$ | $82.4_{\pm0.31}$ | $98.9_{\pm0.10}$ | $71.2_{\pm0.21}$ |
| | specific-relation | $63.3_{\pm0.34}$ | $71.4_{\pm0.32}$ | $82.6_{\pm0.30}$ | $\mathbf{99.5}_{\pm0.06}$ | $64.8_{\pm0.21}$ |
| WN18RR | uncondition | $72.6_{\pm0.35}$ | $74.2_{\pm0.33}$ | $85.2_{\pm0.31}$ | — | $56.4_{\pm0.20}$ |
| | pattern | $\mathbf{77.0}_{\pm0.34}$ | $\mathbf{80.8}_{\pm0.31}$ | $86.8_{\pm0.28}$ | $93.5_{\pm0.24}$ | $\mathbf{83.3}_{\pm0.15}$ |
| | relation-number | $74.0_{\pm0.34}$ | $77.4_{\pm0.31}$ | $86.3_{\pm0.28}$ | $95.3_{\pm0.25}$ | $78.9_{\pm0.20}$ |
| | entity-number | $73.2_{\pm0.37}$ | $77.9_{\pm0.35}$ | $\mathbf{87.2}_{\pm0.33}$ | $85.2_{\pm0.28}$ | $65.1_{\pm0.18}$ |
| | specific-entity | $73.6_{\pm0.38}$ | $75.6_{\pm0.37}$ | $86.2_{\pm0.36}$ | $89.0_{\pm0.31}$ | $65.2_{\pm0.21}$ |
| | specific-relation | $73.0_{\pm0.35}$ | $75.2_{\pm0.33}$ | $85.7_{\pm0.30}$ | $\mathbf{96.1}_{\pm0.19}$ | $60.8_{\pm0.21}$ |
| DBpedia50 | uncondition | $64.3_{\pm0.35}$ | $66.2_{\pm0.33}$ | $79.5_{\pm0.30}$ | — | $51.0_{\pm0.24}$ |
| | pattern | $73.8_{\pm0.37}$ | $76.6_{\pm0.36}$ | $86.8_{\pm0.26}$ | $\mathbf{88.4}_{\pm0.36}$ | $\mathbf{79.2}_{\pm0.20}$ |
| | relation-number | $72.1_{\pm0.32}$ | $76.1_{\pm0.30}$ | $87.5_{\pm0.22}$ | $80.6_{\pm0.43}$ | $79.1_{\pm0.22}$ |
| | entity-number | $\mathbf{75.2}_{\pm0.37}$ | $80.3_{\pm0.35}$ | $92.4_{\pm0.29}$ | $84.0_{\pm0.26}$ | $63.3_{\pm0.22}$ |
| | specific-entity | $73.7_{\pm0.33}$ | $78.7_{\pm0.31}$ | $88.4_{\pm0.35}$ | $79.6_{\pm0.40}$ | $62.9_{\pm0.22}$ |
| | specific-relation | $\mathbf{75.2}_{\pm0.31}$ | $\mathbf{80.6}_{\pm0.29}$ | $\mathbf{93.7}_{\pm0.20}$ | $84.2_{\pm0.36}$ | $60.3_{\pm0.20}$ |

Table 2: Average scores on FB15k237 datasets under five conditions

| Model | Jaccard | Dice | Overlap | Accuracy | Smatch |
|---|---|---|---|---|---|
| GPT-4o + 2-hop subgraph | $2.4_{\pm0.10}$ | $3.1_{\pm0.13}$ | $5.3_{\pm0.20}$ | $77.5_{\pm0.31}$ | $37.9_{\pm0.27}$ |
| Kimi K2 + 2-hop subgraph | $3.1_{\pm0.11}$ | $4.7_{\pm0.17}$ | $8.5_{\pm0.24}$ | $71.6_{\pm0.34}$ | $42.4_{\pm0.22}$ |
| Grok-3 + 2-hop subgraph | $2.5_{\pm0.09}$ | $3.7_{\pm0.12}$ | $6.9_{\pm0.21}$ | $75.6_{\pm0.38}$ | $43.5_{\pm0.21}$ |
| DeepSeek-V3 + 2-hop subgraph | $2.1_{\pm0.09}$ | $2.8_{\pm0.11}$ | $6.3_{\pm0.26}$ | $73.9_{\pm0.33}$ | $41.8_{\pm0.27}$ |
| DeepSeek-V3 + RAG | $5.3_{\pm0.15}$ | $6.7_{\pm0.17}$ | $10.4_{\pm0.46}$ | $76.6_{\pm0.35}$ | $41.8_{\pm0.27}$ |
| GPT5(Thinking) + 2-hop subgraph | $18.7_{\pm0.32}$ | $21.9_{\pm0.35}$ | $37.3_{\pm0.46}$ | $92.8_{\pm0.28}$ | $32.9_{\pm0.27}$ |
| CtrlHGen | $\mathbf{64.3}_{\pm0.33}$ | $\mathbf{71.9}_{\pm0.31}$ | $\mathbf{83.0}_{\pm0.29}$ | $\mathbf{96.6}_{\pm0.84}$ | $\mathbf{73.3}_{\pm0.16}$ |

## 4.2 EXPERIMENT RESULTS AND ANALYSIS

We evaluated the quality and controllability of generated hypotheses on three datasets under five conditions: *pattern*, *relation-number*, *entity-number*, *specific-entity*, and *specific-relation* (see Section 3.3). As baselines, we use AbductiveKGR (Bai et al., 2024b) under unconditional settings (denoted as uncondition) to highlight the improvements of our approach. The results are reported in Table 1. We further compare several advanced LLMs, including GPT-4o Achiam et al. (2023), Kimi K2 (Team et al., 2025), Grok-3 (xAI, 2025), and Deepseek-V3 (Liu et al., 2024a), on FB15k237 dataset under five conditions. For these models, 2-hop subgraphs of observation entities in triple form are included as part of the prompt to compensate for their lack of KG structural knowledge. For all LLMs above, we did not use the thinking mode. And their temperatures are uniformly set to 0.0. In addition, we also added one of the most advanced reasoning models, GPT5, and adopted the thinking mode. At the same time, we constructed an attempt to combine the raw model DeepSeek-V3 with RAG. Average results across five conditions are reported in Table 2, with details provided in Appendix C.1.

Compared to AbductiveKGR (uncondition), our model shows notable improvements in semantic similarity under conditional constraints, with most condition-adherence accuracies exceeding 80%. This improvement likely stems from the additional guidance provided by the control conditions (we further provide a case in Section 4.5 whether the model can handle irrelevant control conditions). Structural conditions generally outperform semantic-focused ones in semantic similarity,

with the fixed-format pattern condition achieving the best results. While both specific-entity and specific-relation conditions similarly enhance semantic similarity, the model shows a clear adherence preference for specific-relation.

On the other hand, the performance of LLMs remains very poor, even on common-sense knowledge graphs. We attribute this issue to two main factors. First, LLMs lack the ability to fully comprehend structured data, while this task requires generating correct structured query graphs rather than merely capturing semantic meaning. Moreover, when the number of observed entities is large, their two-hop subgraphs expand rapidly, producing lengthy textual representations that further challenge the model. Second, the knowledge embedded in LLMs may conflict with that of the knowledge graph. For example, given an observation set containing several singers including Justin Bieber and Kendrick Lamar, Grok-3 classified them as singers who have made hip-hop music, whereas in the knowledge graph, Justin Bieber is not a hip-hop singer. Such contradictions can significantly affect performance on certain domain-specific data. For more analysis, please refer to Appendix C.1.

### 4.3 ABLATION STUDY

We studied the influence of two proposed components of CtrlHGen, dataset augmentation based on sub-logical decomposition and the reward function.

**Sub-logical Decomposition**. We evaluate 13 logical patterns on DBpedia-50 using predefined patterns as conditions. The evaluation is conducted under two settings: one with the data augmentation strategy and one without it. As shown in Fig. 5, sub-logical decomposition significantly improves the Jaccard Index, especially for complex patterns involving disjunctions and negations, while maintaining similar Accuracy between two settings. This indicates that the improvement in long logic is due to the enhanced understanding of the internal logical structure rather than relying on external prompts. Notably, improvements also appear on simple patterns (e.g., 1p), indicating the model benefits from decomposing logic into simpler sub-components.

**Reward Function**. We investigate different reward functions on WN18RR with the "pattern" condition. The results has been shown in Table 3. Reinforcement learning notably improves generalization and reduces accuracy variance compared to supervised learning. Removing Dice and Overlap rewards weakens performance, indicating that Jaccard alone is too strict and may hinder convergence. Excluding the condition-adherence reward slightly improves semantic similarity but harms condition adherence, confirming our reward design effectively balances both objectives. We further analyzed the possible reasons why semantic similarity slightly decreased when conditional adherence was introduced in Appendix C.

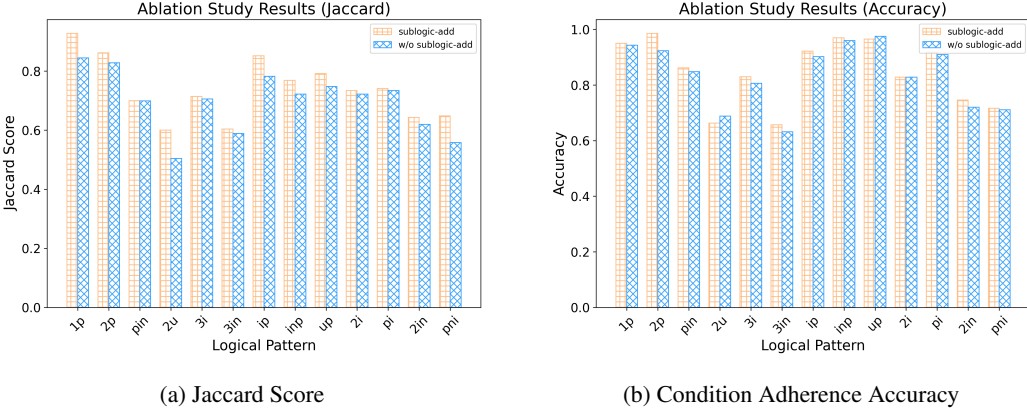

(a) Jaccard Score    (b) Condition Adherence Accuracy

Figure 5: Results of ablation studies for the sub-logical decomposition.

### 4.4 VISUALIZATION

To evaluate controllability, we sampled 100 hypothesis-observation pairs from the FB15k-237 test set for each category defined by the number of relations (1, 2, or 3) in the reference hypothesis.

Table 3: Results of ablation studies for the reward function.

| Model | Semantic Similarity | | | Condition Adherence | | Average |
|---|---|---|---|---|---|---|
| | Jaccard | Dice | Overlap | Accuracy | Smatch | |
| CtrlHG(w/o RL) | $71.5_{\pm 0.37}$ | $75.8_{\pm 0.35}$ | $83.7_{\pm 0.33}$ | $81.5_{\pm 0.38}$ | $79.0_{\pm 0.18}$ | 78.3 |
| CtrlHG(w/o Dice and Overlap) | $74.8_{\pm 0.34}$ | $78.2_{\pm 0.33}$ | $85.1_{\pm 0.30}$ | $90.3_{\pm 0.25}$ | $\underline{82.0}_{\pm 0.15}$ | $\underline{82.1}$ |
| CtrlHG(w/o Condition Adherence) | $\mathbf{77.5}_{\pm 0.33}$ | $\mathbf{81.6}_{\pm 0.31}$ | $\mathbf{87.8}_{\pm 0.29}$ | $68.3_{\pm 0.46}$ | $75.0_{\pm 0.22}$ | 78.0 |
| CtrlHG | $\underline{77.0}_{\pm 0.34}$ | $\underline{80.8}_{\pm 0.31}$ | $\underline{86.8}_{\pm 0.28}$ | $\mathbf{93.5}_{\pm 0.24}$ | $\mathbf{83.3}_{\pm 0.15}$ | $\mathbf{84.3}$ |

We compared the number of predicate relations in generated hypotheses under two settings: with and without relation-number constraints. As shown in Fig. 6, without conditional constraints, the model tends to generate hypotheses with a larger number of predicate relations, making it difficult to generate hypotheses with only one relation. However, when conditional constraints are applied, the majority of generated hypotheses align with the expected number of predicates. This experiment further demonstrates the strong controllability of our model.

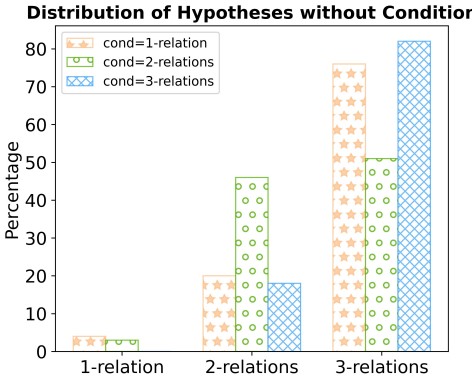
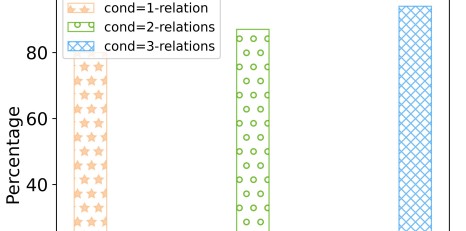

(a) Without condition constraint      (b) With relation-number condition

Figure 6: Visualization of Relation-number Distribution in Generated Hypotheses

## 4.5 CASE STUDY

To demonstrate our controlbility we present two representative cases from FB15k-237, with results provided in Appendix. In the first case (Fig. 8), the observation consists of four music genres: {Blues, Jazz, Rhythm_and_Blues, Bebop}. As the logical pattern conditions grow in complexity, the model produces increasingly fine-grained answers. For instance, under the basic "1p" pattern it identifies their common parent genre, while more complex patterns enable it to retrieve finer details such as artists associated with these genres. In the second case shown in Fig. 9, it focuses on specific entities. For strongly related entities such as Yahoo, the model is able to identify clear connections with the observation set. Even for entities with weaker relationships, such as two movies, the model can still capture hidden associations between them. Surpringsingly, for seemingly unrelated entities like BAFTA_Award_for_Best_Sound, the model is able to generate high-semantic-quality hypotheses by leveraging the logical "or" operator, while still ensuring adherence to the given constraints.

## 5 CONCLUSION

In summary, this paper introduces a new task of controllable abductive reasoning in knowledge graphs to address the limitation of controllability in the existing method. To tackle the challenges when control generating long and complex logical hypotheses, we propose a data augmentation strategy based on sub-logic decomposition, along with smoother semantic and constraint-adherence reward functions. Experimental results demonstrate that our approach significantly improves the controllability and overall quality of the generated hypotheses.

## 6 ACKNOWLEDGEMENTS

The corresponding author is Yangqiu Song. We owe sincerely thanks to all authors for their valuable efforts and contributions. The authors of this paper are supported by the ITSP Platform Research Project (ITS/189/23FP) from ITC of Hong Kong, SAR, China, and the AoE (AoE/E-601/24-N), the RIF (R6021-20) and the GRF (16205322) from RGC of Hong Kong, SAR, China and the National Natural Science Foundation of China (No.62462007 and No.62302023). We also thank the support of RGC JRFS2526-6S10.

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

## A    DETAILS FOR OBSERVATION-HYPOTHESIS SAMPLE

Given a knowledge graph $G$ and a predefined logical pattern $P$, the algorithm begins by sampling a random node $v$ and recursively constructs a hypothesis such that $v$ is one of its conclusions and the hypothesis conforms to the logical type specified by $P$. During the recursive process, the algorithm examines the current operation in the hypothesis structure. If the operation is projection, the algorithm randomly selects an incoming edge $(u, r, v)$ of node $v$, then recursively generates a sub-hypothesis rooted at node $u$ according to the corresponding subtype of $P$. If the operation is intersection, the algorithm recursively constructs sub-hypotheses using the same node $v$ for each subtype, since all sub-hypotheses must conclude with $v$. If the operation is union, it applies the recursion to one subtype using node $v$, and to the remaining subtypes using randomly selected nodes. This is because, under union, only one of the sub-hypotheses needs to have $v$ as its conclusion.

For the sub-logic decomposition, we decompose a hypothesis into its sub-logical hypothesis $H_{sub}$ based on the type of reference hypothesis $H$. For example, a logical pattern "inp" can be decomposed into two sublogical patterns "2p". Then we calculate the corresponding conclusions of these two "2p" logical hypotheses respectively as sub-observations, thereby constructing the sub-logic observation-hypothesis set.

## B    MORE EXPERIMENT DETAILS

For all experiments, we set the learning rate to 1e-5 and use a batch size of 256 during supervised training. The supervised training process consists of two stages. In the first stage, the model is trained for 400 epochs, including a 50-epoch warm-up phase. In the second stage, which involves conditional supervised training, we train for 50 epochs with a 5-epoch warm-up. For reinforcement learning, a smaller batch size of 32 is used, and each group samples 4 candidate answers. The hyperparameters $\lambda_1$, $\lambda_2$, and $\lambda_3$ are set to 1.0, 0.5, and 0.5, respectively. And then we set $\alpha = 0.5$.

### B.1    SMATCH

Smatch (Cai & Knight, 2013) is an evaluation metric for Abstract Meaning Representation (AMR) graphs, which are directed acyclic graphs with two node types (variable and concept) and three edge types (instance, attribute, and relation). Given a predicted graph $G_p$ and a gold graph $G_g$, Smatch$(G_p, G_g)$ is computed by finding an approximately optimal mapping between the variable nodes of the two graphs and matching their edges. Following the settings of Bai et al. (2024b), we transform the hypothesis graph $G(H)$ into an AMR graph $GA(H)$ by adding virtual nodes and instance edges, and then calculate Smatch. In short, Smatch is used to measure the degree of similarity between the generated hypothesis and the ground truth in the test set.

## C    MORE RESULTS

### C.1    DETAILED RESULTS

Here, we reported our detailed results of LLMs' performance in Table 4. We also showed our prompts in Fig 7. We found that large language models are sometimes greatly influenced by semantics, thus neglecting the role of correct structure. For example, when the observation is {Librarian, Lawyer, Mathematician, Physicist, Scientist-GB}, Grok-3 will answer whether they are working in a library or in a law-related profession. However, the correct query for reference is the occupation of Gottfrie Wilhelm von Leibniz or the occupation of those influenced by Italo Calvino. In this example, the large language model found the most relevant semantic content but ignored that they could not meet all situations. Even more strangely, large language models sometimes include the entities within observations in the hypotheses they generate. Given an observation {Fever, Fatigue, Headache}, LLMs did not find any drugs that could treat them or diseases with these three symptoms. Instead, it included these three observed entities and predicates belonging to a certain symptom in its logical assumptions. That is, these observations are symptoms of fever, Headache and Fatigue. We believe this is because the large language model has not fully understood the structural relationship, thus confusing the contents of the input edge and the output edge.

---

**Input:** Observation, Logic Patterns, 2-hop subgraph, condition

**[Format Specification]:**  'answer: <hypothesis>'.
**[Instructions]:**  Do not add quotes, explanations, or extra text and only replace <hypothesis> with your generated content.
**[Task]:**  Now you need to do the abductive reasoning in knowledge graphs. The observation (consist of entity id and semantic) is <observation>.
Your task is to generate the hypothesis whose form is first order logic in <logic patterns>, where i means intersection, u means union, n means negation. p denotes the relation and e is the entity.
Here is the related 2-hop subgraph <2-hop subgraph> for you, which may help you. For each (u,v,k), u is the source node, v is the target node, k is the relation. Each form is 'id: Semantic content'.
Now generate the hypothesis, with the format in the <logic patterns> . Please note that you need to make sure the hypothesis you generate satisfy the <condition> .

---

Figure 7: Prompt Example.

On the one hand, we found that GPT5(Thinking) has achieved a significant performance improvement Firstly, the model can follow the control conditions in most cases. Secondly, higher semantic similarity is achieved under all five conditions. In contrast, models are more likely to generate hypotheses with higher semantic similarities under the control of semantic content than under structural control. This might be because the model itself is better at capturing based on semantics compared to structured reasoning. However, they still have a considerable gap compared to CtrlHGen, indicating that abductive reasoning tasks with structured knowledge remain challenging for advanced large language models.

On the other hand, Deepseek-V3 with RAG has improved performance under the condition of semantic control, but the results remains almost unchanged under the condition of structural control. We believe this can be attributed to two primary reasons: First, RAG primarily enhances semantic retrieval, enabling the model to fetch more semantically relevant context. It offers limited benefit when precise structural constraints are imposed, as these require strict path conformance rather than mere semantic relevance. Second, the provided 2-hop subgraph already serves as a highly informative prompt. Since the depth of all 13 predefined logical patterns is 2, this 2-hop subgraph covers most of the structural information required for hypothesis generation.

The consistently poor performance under structural control instead reveals the models' persistent weakness in complex structural reasoning over graphs. Compared to standard KGQA, which only requires interpreting and following one given logical chain, abductive reasoning is fundamentally more challenging: it demands that the model simultaneously consider all relevant logical chains surrounding a set of observed entities and abduce the single most explanatory multi-hop hypothesis. This inverse, open-ended search process imposes significantly greater demands on structural understanding and logical synthesis, an area where current LLMs still fall short.

We also compared the experimental results of GPT5 (thinking) under different temperature settings on FB15k237 dataset under the 'patttern' condition. The results are shown in Table 5. We found that a temperature of 0.0 can ensure a balance between semantic similarity and condition adherence. Excessively high temperatures may enhance the ability to explore, thereby improving semantic similarity, but they will significantly reduce condition adherence.

## C.2  ANALYSIS FOR CONDITION ADHERENCE REWARD

We have further analyzed the ablation study presented in the paper, comparing the performance of each logical pattern with and without the Condition Adherence(CA) reward. The results are summarized in Table 6 and Table 7.

Table 4: The results of controllable abductive reasoning under different conditions. (Result: average score $\pm$ standard deviation.)

| Dataset | Condition | Semantic Similarity | | | Condition Adherence | |
|---|---|---|---|---|---|---|
| | | Jaccard | Dice | Overlap | Accuracy | Smatch |
| GPT-4o + 2-hop subgraph | pattern | $4.7_{\pm 0.19}$ | $5.1_{\pm 0.20}$ | $7.7_{\pm 0.26}$ | $85.3_{\pm 0.18}$ | $55.6_{\pm 0.29}$ |
| | relation-number | $1.9_{\pm 0.08}$ | $2.8_{\pm 0.11}$ | $5.4_{\pm 0.19}$ | $74.4_{\pm 0.49}$ | $44.1_{\pm 0.26}$ |
| | entity-number | $2.2_{\pm 0.08}$ | $3.3_{\pm 0.12}$ | $6.0_{\pm 0.20}$ | $84.3_{\pm 0.36}$ | $45.3_{\pm 0.22}$ |
| | specific-entity | $2.5_{\pm 0.12}$ | $3.2_{\pm 0.14}$ | $5.0_{\pm 0.21}$ | $77.8_{\pm 0.24}$ | $20.7_{\pm 0.27}$ |
| | specific-relation | $0.9_{\pm 0.06}$ | $1.3_{\pm 0.08}$ | $2.4_{\pm 0.13}$ | $65.5_{\pm 0.26}$ | $23.8_{\pm 0.23}$ |
| Kimi K2 + 2-hop subgraph | pattern | $3.1_{\pm 0.10}$ | $4.6_{\pm 0.14}$ | $7.7_{\pm 0.22}$ | $82.4_{\pm 0.33}$ | $50.2_{\pm 0.21}$ |
| | relation-number | $2.4_{\pm 0.09}$ | $3.6_{\pm 0.12}$ | $8.5_{\pm 0.26}$ | $71.1_{\pm 0.49}$ | $47.0_{\pm 0.19}$ |
| | entity-number | $2.2_{\pm 0.09}$ | $3.2_{\pm 0.12}$ | $6.0_{\pm 0.20}$ | $62.3_{\pm 0.41}$ | $35.8_{\pm 0.19}$ |
| | specific-entity | $4.2_{\pm 0.18}$ | $5.7_{\pm 0.20}$ | $10.8_{\pm 0.26}$ | $69.0_{\pm 0.30}$ | $38.4_{\pm 0.25}$ |
| | specific-relation | $3.6_{\pm 0.11}$ | $5.2_{\pm 0.15}$ | $9.5_{\pm 0.26}$ | $73.4_{\pm 0.19}$ | $40.5_{\pm 0.24}$ |
| Grok-3 + 2-hop subgraph | pattern | $3.8_{\pm 0.11}$ | $5.7_{\pm 0.15}$ | $12.0_{\pm 0.28}$ | $83.0_{\pm 0.37}$ | $61.2_{\pm 0.24}$ |
| | relation-number | $1.8_{\pm 0.07}$ | $2.8_{\pm 0.10}$ | $4.6_{\pm 0.17}$ | $70.5_{\pm 0.45}$ | $40.0_{\pm 0.26}$ |
| | entity-number | $1.9_{\pm 0.07}$ | $2.8_{\pm 0.11}$ | $4.9_{\pm 0.18}$ | $70.9_{\pm 0.45}$ | $42.2_{\pm 0.23}$ |
| | specific-entity | $2.7_{\pm 0.12}$ | $3.7_{\pm 0.14}$ | $6.0_{\pm 0.22}$ | $76.3_{\pm 0.31}$ | $38.8_{\pm 0.27}$ |
| | specific-relation | $2.3_{\pm 0.08}$ | $3.4_{\pm 0.11}$ | $7.2_{\pm 0.22}$ | $77.2_{\pm 0.32}$ | $35.4_{\pm 0.27}$ |
| Deepseek-V3 + 2-hop subgraph | pattern | $2.7_{\pm 0.10}$ | $4.0_{\pm 0.13}$ | $7.1_{\pm 0.21}$ | $79.1_{\pm 0.50}$ | $47.2_{\pm 0.31}$ |
| | relation-number | $0.9_{\pm 0.04}$ | $1.3_{\pm 0.06}$ | $5.5_{\pm 0.22}$ | $70.6_{\pm 0.31}$ | $39.1_{\pm 0.32}$ |
| | entity-number | $1.3_{\pm 0.08}$ | $1.7_{\pm 0.10}$ | $3.4_{\pm 0.16}$ | $69.2_{\pm 0.29}$ | $40.3_{\pm 0.22}$ |
| | specific-entity | $3.7_{\pm 0.15}$ | $4.7_{\pm 0.17}$ | $10.2_{\pm 0.29}$ | $75.8_{\pm 0.30}$ | $41.6_{\pm 0.28}$ |
| | specific-relation | $1.7_{\pm 0.09}$ | $2.3_{\pm 0.11}$ | $5.4_{\pm 0.20}$ | $74.2_{\pm 0.28}$ | $40.6_{\pm 0.23}$ |
| GPT5(Thinking)+2-hop subgraph | pattern | $14.8_{\pm 0.30}$ | $17.4_{\pm 0.32}$ | $30.6_{\pm 0.42}$ | $83.8_{\pm 0.37}$ | $71.5_{\pm 0.31}$ |
| | relation-number | $14.6_{\pm 0.30}$ | $17.1_{\pm 0.32}$ | $31.5_{\pm 0.44}$ | $96.6_{\pm 0.17}$ | $56.8_{\pm 0.17}$ |
| | entity-number | $17.8_{\pm 0.30}$ | $22.0_{\pm 0.33}$ | $44.9_{\pm 0.46}$ | $95.3_{\pm 0.21}$ | $54.2_{\pm 0.19}$ |
| | specific-entity | $24.1_{\pm 0.36}$ | $27.4_{\pm 0.39}$ | $40.1_{\pm 0.49}$ | $94.2_{\pm 0.26}$ | $31.9_{\pm 0.21}$ |
| | specific-relation | $22.1_{\pm 0.34}$ | $25.8_{\pm 0.37}$ | $39.6_{\pm 0.46}$ | $94.5_{\pm 0.22}$ | $28.1_{\pm 0.20}$ |
| Deepseek-V3 + RAG | pattern | $2.8_{\pm 0.09}$ | $3.8_{\pm 0.22}$ | $6.1_{\pm 0.21}$ | $78.5_{\pm 0.34}$ | $48.2_{\pm 0.35}$ |
| | relation-number | $1.6_{\pm 0.08}$ | $2.3_{\pm 0.11}$ | $3.8_{\pm 0.19}$ | $69.3_{\pm 0.40}$ | $34.5_{\pm 0.26}$ |
| | entity-number | $0.8_{\pm 0.04}$ | $1.4_{\pm 0.07}$ | $3.8_{\pm 0.19}$ | $72.3_{\pm 0.40}$ | $39.2_{\pm 0.24}$ |
| | specific-entity | $13.7_{\pm 0.31}$ | $15.4_{\pm 0.33}$ | $23.0_{\pm 0.42}$ | $82.8_{\pm 0.41}$ | $25.9_{\pm 0.24}$ |
| | specific-relation | $7.6_{\pm 0.27}$ | $10.5_{\pm 0.30}$ | $15.4_{\pm 0.36}$ | $80.2_{\pm 0.30}$ | $16.8_{\pm 0.26}$ |

Table 5: Temperature sensitivity experiment.

| Temperature | Semantic Similarity | | | Condition Adherence | | Average |
|---|---|---|---|---|---|---|
| | Jaccard | Dice | Overlap | Accuracy | Smatch | |
| t=1.0 | $15.8_{\pm 0.31}$ | $18.2_{\pm 0.33}$ | $28.5_{\pm 0.41}$ | $75.3_{\pm 0.43}$ | $68.4_{\pm 0.18}$ | 41.2 |
| t=0.5 | $13.4_{\pm 0.28}$ | $15.8_{\pm 0.31}$ | $28.8_{\pm 0.41}$ | $79.2_{\pm 0.40}$ | $69.1_{\pm 0.18}$ | 41.2 |
| t=0.0 | $14.8_{\pm 0.30}$ | $17.4_{\pm 0.32}$ | $30.6_{\pm 0.42}$ | $83.8_{\pm 0.37}$ | $71.5_{\pm 0.31}$ | 43.6 |

- For logical patterns involving negation (such as 2in, pin, and inp), we observed that even without conditional adherence rewards, the model is often able to identify the correct logical structure on its own and achieve higher accuracy. In these cases, enforcing constraint adherence may overly limit the model's exploratory flexibility, leading to suboptimal semantic performance.

- In contrast, for logical patterns that involve only intersection (such as pi, ip, 2i, 3i), we found a strong correlation between improved constraint satisfaction and enhanced semantic similarity. Without reinforcement signals guiding the model to comply with constraints, it tends to generate alternative formats that deviate from the intended structure, resulting in decreased semantic quality.

- Interestingly, for the '3in' pattern, the model appears to strike a balance between intersection and negation: regardless of whether constraints are enforced, the resulting hypotheses exhibit comparable semantic similarity.

Table 6: Ablation Results of Jaccard Score

| logical pattern | 2in | pin | inp | pni | up | 2u | 3in | 1p | 2p | pi | ip | 2i | 3i |
|---|---|---|---|---|---|---|---|---|---|---|---|---|---|
| CtrlHGen(w/o RL) | 63.6 | 68.1 | 67.6 | 65.2 | 67.0 | 81.9 | 69.2 | 75.4 | 79.3 | 72.6 | 73.8 | 70.1 | 74.8 |
| CtrlHGen(w/o CA) | 76.2 | 75.9 | 72.3 | 71.1 | 71.6 | 85.3 | 70.4 | 91.2 | 85.1 | 75.8 | 82.1 | 78.9 | 71.9 |
| CtrlHGen | 71.7 | 73.2 | 69.7 | 69.1 | 70.3 | 84.9 | 70.2 | 91.3 | 85.3 | 76.4 | 82.8 | 79.8 | 77.2 |
| Difference | -4.5 | -2.7 | -2.6 | -2.0 | -1.3 | -0.4 | -0.2 | 0.1 | 0.2 | 0.6 | 0.7 | 0.9 | 5.3 |

Table 7: Ablation Results of Condition Adherence Accuracy

| logical pattern | 2in | pin | inp | pni | up | 2u | 3in | 1p | 2p | pi | ip | 2i | 3i |
|---|---|---|---|---|---|---|---|---|---|---|---|---|---|
| CtrlHGen (w/o RL) | 58.7 | 60.1 | 98.2 | 78.7 | 98.5 | 93.9 | 93.2 | 45.5 | 84.6 | 88.5 | 91.4 | 96.7 | 70.6 |
| CtrlHGen (w/o CA) | 82.4 | 84.7 | 78.7 | 79.1 | 91.6 | 97.0 | 34.2 | 65.9 | 74.8 | 57.0 | 58.4 | 76.9 | 16.5 |
| CtrlHGen | 84.5 | 98.6 | 84.4 | 85.8 | 98.7 | 96.3 | 95.4 | 89.0 | 96.4 | 98.2 | 98.3 | 98.6 | 90.1 |

## C.3 MORE BASELINES

Here, we incorporated the data augmentation strategy proposed in Logic-Gen (Asai & Hajishirzi, 2020) as an additional baseline. We also compared it with our method CtrlHGen and AbductiveKGR without data augmentation but only by introducing conditional tokens. Since these two methods don't employ reinforcement learning for conditional control, we report results after supervised training, ensuring a fair comparison. We conducted experiments on the DBpedia50 dataset and selected 'pattern' and 'specific-relation' respectively to represent structural control and semantic control. The results are reported in Table 8 and 9.

The experiments reveal that, while Logic-Gen's data augmentation indeed improves the model's overall grasp of logical patterns, it remains inferior to our sub-logic decomposition approach. We believe this is because the sub-logic decomposition forces the model to deeply understand and compose longer, more intricate logical chains step-by-step, leading to substantially stronger reasoning capability on complex hypotheses. It more effectively mitigates hypothesis space collapse, thereby significantly enhancing compliance when strict structural conditions are imposed.

Table 8: Results on DBpedia50 dataset under the 'pattern' condtion.

| Model | Semantic Similarity | | | Condition Adherence | |
|---|---|---|---|---|---|
| | Jaccard | Dice | Overlap | Accuracy | Smatch |
| AbductiveKGR+condition token | $68.2_{\pm 0.34}$ | $73.2_{\pm 0.32}$ | $80.6_{\pm 0.29}$ | $66.6_{\pm 0.47}$ | $77.5_{\pm 0.20}$ |
| Logic-Gen | $69.5_{\pm 0.34}$ | $73.5_{\pm 0.32}$ | $79.9_{\pm 0.30}$ | $65.9_{\pm 0.47}$ | $77.5_{\pm 0.21}$ |
| CtrlHGen | $70.1_{\pm 0.33}$ | $74.0_{\pm 0.31}$ | $80.8_{\pm 0.29}$ | $73.1_{\pm 0.41}$ | $80.8_{\pm 0.17}$ |

## C.4 CASE STUDY

In this section, we show the results of two case study in Fig 8 and Fig 9.

## C.5 MULTI-DIALOGUE CASE

In this section, we implemented a simple yet highly interactive multi-round dialogue system that automatically adjusted control conditions based on the user's evolving intentions and the outcomes of previous rounds. We leveraged a large language model (DeepSeek-V3) to intelligently select appropriate control conditions according to the user's expressed intent. The prompt used for this condition-selection LLM is presented in Fig. 10. At each turn, the LLM generated updated control

Table 9: Results on the DBpedia50 dataset under the 'specific-relation' condition.

| Model | Semantic Similarity | | | Condition Adherence | |
|---|---|---|---|---|---|
| | Jaccard | Dice | Overlap | Accuracy | Smatch |
| AbductiveKGR + condition token | $69.3_{\pm 0.35}$ | $73.0_{\pm 0.33}$ | $86.4_{\pm 0.31}$ | $78.6_{\pm 0.40}$ | $58.0_{\pm 0.23}$ |
| Logic-Gen | $70.4_{\pm 0.35}$ | $73.4_{\pm 0.33}$ | $88.0_{\pm 0.29}$ | $75.9_{\pm 0.42}$ | $54.9_{\pm 0.23}$ |
| CtrlHGen | $72.7_{\pm 0.33}$ | $77.2_{\pm 0.31}$ | $90.7_{\pm 0.27}$ | $80.0_{\pm 0.40}$ | $51.6_{\pm 0.23}$ |

---

**Observation:** Blues, Jazz, Rhythm_and_blues, Bebop

**Condition1:** Logical Pattern 1p
**Hypothesis 1:** $H = V_? : Parent\_genre(Hard\_bop, V_?)$
**Interpretations 1:** The music genre that originates from the Hard_bop genre.
**Conclusion 1:** Blues, Jazz, Rhythm_and_blues, Bebop.
**Jaccard Score:** 1.0

---

**Condition2:** Logical Pattern 2p
**Hypothesis 2:** $H = V_? : Parent\_genre(P_?, V_?) \land genre(McCoy\_Tyner, P_?)$
**Interpretations 2:** The musical genre that originates from the genre which is associated with the artist McCoy_Tyner.
**Conclusion 2:** Blues, Jazz, Rhythm_and_blues, Bebop.
**Jaccard Score:** 1.0

---

**Condition3:** Logical Pattern ip
**Hypothesis 3:** $H = V_? : Parent\_genre(Hard\_bop, V_?) \land genre(Roy\_Haynes, P_?) \land genre(McCoy\_Tyner, P_?)$
**Interpretations 3:** The musical genre that originates from the Hard_bop genre and is associated with the artist Roy_Haynes and McCoy_Tyner.
**Conclusion 3:** Blues, Jazz, Rhythm_and_blues, Bebop.
**Jaccard Score:** 1.0

---

Figure 8: Case study of Logic Control.

conditions by jointly considering the hypothesis produced in the previous round, its derived conclusions, the corresponding Jaccard similarity score, and the current user input. These dynamically selected conditions were then passed back to the core hypothesis generation model. A complete interaction example is shown in Fig. 11.

In this case, the initial observation consisted of four songs. In the first round, the user expressed interest in connections related to the acoustic guitar. The system accordingly generated a relatively broad hypothesis that slightly over-covered the observed entities. In the second round, the user asked who the artist was; the LLM selected "specific-relation" as the control condition to focus the generation. Although a relevant hypothesis was produced, it remained somewhat vague. Consequently, in the third round, the user requested a simpler logical structure. The LLM responded by enforcing the simplest available logic pattern, successfully revealing that all four songs were authored by Tracy Lawrence. Finally, wishing to explore the observation more deeply, the user sought additional related information. The LLM then imposed a relation count of three as the control condition, prompting the model to generate a richer, more complex hypothesis that incorporated two different associated artists.

Through this multi-round interaction, the system seamlessly combines structural and semantic control signals, gradually improving the relevant hypotheses closely related to the user's constantly evolving exploration goals. It demonstrates the potential of our method in real-world scenarios.

Observation: Avid_Technology, Intel_Corporation, IBM, Microsoft_Corporation, Seattle_Mariners, Walmart, Cisco_Systems, Inc., Golden_State_Warriors, Seattle_Supersonics, Los_Angeles_Kings, San_Jose_Sharks, Pixar, Amazon.com, eBay

**Condition 1:** Entity contains Smokin_Aces
**Hypothesis 1:** $H = V_? : Employ(V_?, P_?) \wedge Film\_crew\_role(Smokin\_Aces, P_?)$
**Interpretations 1:** The organization/company that employs people who have served as a crew member in the film Smokin_Aces.
**Conclusion 1:** Avid_Technology, Intel_Corporation, IBM, Microsoft_Corporation, Seattle_Mariners, Walmart, Cisco_Systems, Inc., Golden_State_Warriors, Seattle_Supersonics, Los_Angeles_Kings, San_Jose_Sharks, Pixar, Amazon.com, eBay.
**Jaccard Score:** 1.0

**Condition 2:** Entity contains The_Phantom
**Hypothesis 2:** $H = V_? : Employ(V_?, P_?) \wedge Film\_crew\_role(The\_Phantom, P_?)$
**Interpretations 2:** The organization/company that employs people who have served as a crew member in the film The_Phantom.
**Conclusion 2:** Avid_Technology, Intel_Corporation, IBM, Microsoft_Corporation, Seattle_Mariners, Walmart, Cisco_Systems, Inc., Golden_State_Warriors, Seattle_Supersonics, Los_Angeles_Kings, San_Jose_Sharks, Pixar, Amazon.com, eBay.
**Jaccard Score:** 1.0

**Condition 3:** Entity contains Yahoo
**Hypothesis 3:** $H = V_? : Employ(V_?, P_?) \wedge \neg Employed(P_?, Yahoo) \wedge Employed(P_?, Avid\_Technology)$
**Interpretations 3:** The organization/company that employs people who have never been employed by Yahoo, but have been employed by Avid_Technology.
**Conclusion 3:** Avid_Technology, Intel_Corporation, IBM, Microsoft_Corporation, Seattle_Mariners, Walmart, Cisco_Systems, Inc., Golden_State_Warriors, Seattle_Supersonics, Los_Angeles_Kings, San_Jose_Sharks, Pixar, Amazon.com, eBay.
**Jaccard Score:** 1.0

**Condition 4:** Entity contains BAFTA_Award_for_Best_Sound
**Hypothesis 4:**
$H = V_? : Employ(V_?, P_?) \wedge Nominated\_for(P_?, BAFTA\_Award\_for\_Best\_Sound)$
$\vee Employed(P_?, Los\_Angeles\_Kings)$
**Interpretations 4:** The organization/company that employs people who have been nominated for BAFTA_Award_for_Best_Sound or have been employed by Los_Angeles_Kings.
**Conclusion 4:** Avid_Technology, Intel_Corporation, IBM, Microsoft_Corporation, Seattle_Mariners, Walmart, Cisco_Systems, Inc., Golden_State_Warriors, Seattle_Supersonics, Los_Angeles_Kings, San_Jose_Sharks, Pixar, Amazon.com, eBay.
**Jaccard Score:** 1.0

Figure 9: Case Study of Entity Semantic Control.

# D    THE USE OF LLMS

In this paper, large language models (LLMs) were employed exclusively for language refinement, such as improving grammar, clarity, and readability of the manuscript. They were not utilized in any stage of the research process itself, including the formulation of ideas, experimental design, data collection, analysis, or interpretation of results.

Input: Observation, Hypothesis(last round), Condition(last round),Jaccard_score(last round), Intention, Logic Patterns

**[Instructions]:** Do not add quotes, explanations, or extra text and only replace <hypothesis> with your generated content.

**[Task]:** Now I am doing the abductive reasoning in knowledge graphs. The observation (consist of entity id: semantic) is Observation. I have generated a hypothesis <Hypothesis> under the condition <Condition>. The conclusion of this hypothesis is <conclusion>. And the jaccard score between the conclusion and the observation is <Jaccard_score>. My intention is that <Intention>. Now your task is to adjust the condition and my model will generate a new hypothesis under the condition to make the jaccard score between the conclusion and the observation higher and conforms to my intention.

**[Format Specification]:** The condition can be one of them 'pattern', 'entitynumber', 'entity', 'relation', 'relationnumber'. For the pattern, you can specify the corresponding first-order logical hypothesis format including <Logic Patterns>. If you choose this condition, your response must be condition: <pattern: <pattern_name>. For the entitynumber, you can specify the number of entities in the hypothesis. If you choose this condition, your response must be condition: <entitynumber: <number>. For the entity, you can specify the entities in the hypothesis. If you choose this condition, your response must be condition: <entity: <entity_name>. For the relation, you can specify the relations in the hypothesis. If you choose this condition, your response must be condition: <relation: <relation_name>. For the relationnumber, you can specify the number of relations in the hypothesis. If you choose this condition, your response must be condition: <relationnumber: <number>.

Figure 10: Prompt for LLM.

Observation: Alibis_(song), I_See_It_Now_(song), If_the_Good_Die_Young, Somebody_Paints_the_Wall

**Intention 1:** I want to know about the connection between these songs and the guitar.
**Condition 1:** Entity contains Acoustic_guitar.
**Hypothesis 1:** $H = V_? : WorkofArtist(V_?, P_?) \land Instrumentof(Acoustic\_guitar, P_?)$
**Interpretations 1:** The songs that are the work of the artist whose instrument is Acoustic_guitar.
**Conclusion 1:** Alibis_(song), I_See_It_Now_(song), If_the_Good_Die_Young, Somebody_Paints_the_Wall, If_I_Had_Any_Pride_Left_at_All.
**Jaccard Score:** 0.66

---

**Intention 2:** I want to know which artist is the author of these songs.
**Condition 2:** Relation contains WorkofArtist.
**Hypothesis 2:** $H = V_? : WorkofArtist(V_?, P_?) \land AssociateArtist(Ken\_Mellons, P_?)$
**Interpretations 2:** The songs that are the work of the artist whose associateArtist is Ken_Mellons.
**Conclusion 2:** Alibis_(song), I_See_It_Now_(song), If_the_Good_Die_Young, Somebody_Paints_the_Wall.
**Jaccard Score:** 1.0

---

**Intention 3:** This is too complex. I want to make the logic simpler.
**Condition 3:** Logic Pattern 1p.
**Hypothesis 3:** $H = V_? : WorkofArtist(V_?, Tracy\_Lawrence)$
**Interpretations 3:** The songs that are the work of the artist Tracy_Lawrence.
**Conclusion 3:** Alibis_(song), I_See_It_Now_(song), If_the_Good_Die_Young, Somebody_Paints_the_Wall.
**Jaccard Score:** 1.0

---

**Intention 4:** I want to know more. Let's explore more.
**Condition 4:** Logic contains 3 relations.
**Hypothesis 4:** $H = V_? : Musicband(V_?, P_?) \land AssociateArtist(Josh\_Logan\_(country\_singer), P_?) \land AssociateArtist(Ken\_Mellons, P_?)$
**Interpretations 4:** The songs that are played by the musical band whose associate artists are Josh_Logan_(country_singer) and Ken_Mellons.
**Conclusion 4:** Alibis_(song), I_See_It_Now_(song), If_the_Good_Die_Young, Somebody_Paints_the_Wall.
**Jaccard Score:** 1.0

Figure 11: Case Study of Multi-round Dialogue.

