# OpenReview forum: "Controllable Logical Hypothesis Generation for Abductive Reasoning in Knowledge Graphs"
_ICLR.cc/2026/Conference — ICLR 2026 Poster_

### Official Review · Reviewer_wmXZ · 2025-10-30

**Soundness:** 3
**Presentation:** 3
**Contribution:** 3
**Rating:** 8
**Confidence:** 4

**Summary:**

Abductive reasoning in knowledge graphs facilitates the creation of plausible logical hypotheses to explain observed entities. This article presents a novel and controllable abductive reasoning approach to overcome the limitations of existing methods, which often lack flexibility in real-world applications. To address challenges including overly lengthy hypotheses due to hypothesis space collapse and the high sensitivity of generated outputs, the authors introduce a sub-logic decomposition-based data augmentation technique, combined with meticulously crafted reinforcement learning reward signals. Comprehensive experiments show that the proposed CtrlHGen method effectively resolves these issues, significantly improving the controllability and semantic quality of hypothesis generation.

**Strengths:**

1.The controllable abductive reasoning approach for knowledge graphs introduced in this article is both innovative and practically significant. By generating plausible hypotheses tailored to user-specific interests, this method holds substantial value for applications such as recommendation systems and scientific discovery.

2.The writing of this paper is clear and easy to understand. This paper naturally puts forward the significance of controllability, the challenges of achieving controllability and how to solve these challenges.

3.The paper presents convincing experimental results. Evaluations across five distinct settings convincingly demonstrate the proposed method's ability to achieve controllability. Furthermore, ablation studies robustly validate the effectiveness and contributions of the approach.

4.The case studies reveal unexpectedly robust results. When presented with control conditions unrelated to the observed entities, the model adeptly introduces logical disjunctions (OR) to concurrently meet both the control constraints and maintain semantic interpretability. This behavior is both intriguing and highly significant.

**Weaknesses:**

1.The author did not explore whether some naive methods could be applied to controllable abductive reasoning tasks. For instance, models based on link prediction in KG, or search methods based on subgraph matching.

2.It seems to me that the poor performance of LLMs is very interesting. But the authors should provide a specific case, including the 2-hop subgraph obtained by the large model and its generation results, so as to more clearly demonstrate the reasons for the poor performance of the large model.

3.Following W1 and W2, the poor performance of LLMs may be limited by some naive searches. The author should make an analysis on this point. For example, whether the 2-hop subgraph can contain valid answers.

**Questions:**

My questions have already been listed in Weakness.

---

> ### Author Response · Authors · 2025-11-19
>
> Dear Reviewer wmXZ,
>
> We sincerely thank the reviewer for the detailed comments and insightful questions. Our response to your comments one by one as follows.
>
> For further experiments and detailed analysis involving large language models, we have to add them in Section 4.2, Appendix C.1 and C.3. All content newly added during the rebuttal phase is highlighted in blue for easy identification.
>
> ---
>
> > W1: The author did not explore whether some naive methods could be applied to controllable abductive reasoning tasks. For instance, models based on link prediction in KG, or search methods based on subgraph matching.
> >
>
> Reply to W1: Thank you for your valuable suggestions. Search-based methods rely heavily on the completeness of the knowledge graph. ***The computational complexity is very high, making them unsuitable for large-scale knowledge graphs.*** Even when control constraints are applied, the search space remains vast. Detailed experimental results and analyses have already been provided in [1].
> ***As for link prediction methods, they are limited to handling relatively simple hypothetical logic.*** When dealing with more complex hypotheses, it becomes challenging to trace back through a larger number of nodes. To our knowledge, the existing reasoning work is limited to multi-hop reasoning using the link prediction method, but cannot handle reverse problems.
>
> [1] Advancing Abductive Reasoning in Knowledge Graphs through Complex Logical Hypothesis Generation.

---

> ### Author Response · Authors · 2025-11-19
>
> > W2: It seems to me that the poor performance of LLMs is very interesting. But the authors should provide a specific case, including the 2-hop subgraph obtained by the large model and its generation results, so as to more clearly demonstrate the reasons for the poor performance of the large model.
> >
>
> > W3: Following W1 and W2, the poor performance of LLMs may be limited by some naive searches. The author should make an analysis on this point. For example, whether the 2-hop subgraph can contain valid answers.
> >
>
> Reply to W2 and W3: Thank you for your insightful review. Here, we provided more detailed experiments and analyses. Specifically, we have added one of the most advanced reasoning models, GPT5, and adopted the thinking mode. At the same time, we constructed an attempt to combine the raw model DeepSeek-V3 with RAG. The results have shown in Table 1 and Table 2.
>
> - ***On the one hand, we found that GPT5(Thinking) has achieved a significant performance improvement.*** Firstly, the model can follow the control conditions in most cases. Secondly, higher semantic similarity is achieved under all five conditions. In contrast, models are more likely to generate  hypotheses with higher semantic similarities under the control of semantic content than under structural control. This might be because the model itself is better at capturing based on semantics compared to structured reasoning. ***However, they still have a considerable gap compared to CtrlHGen, indicating that abductive reasoning tasks with structured knowledge remain challenging for advanced large language models.***
> - ***On the other hand, Deepseek-V3 with RAG  has improved performance under the condition of semantic control, but the results remains almost unchanged under the condition of structural control.*** We believe this can be attributed to two primary reasons: First, RAG primarily enhances semantic retrieval, enabling the model to fetch more semantically relevant context.  It offers limited benefit when precise structural constraints are imposed, as these require strict path conformance rather than mere semantic relevance.  Second, ***the provided 2-hop subgraph already serves as a highly informative prompt. Since the depth of all 13 predefined logical patterns is 2, this 2-hop subgraph covers most of the structural information required for hypothesis generation***.
> - The consistently poor performance under structural control instead reveals the models’ persistent weakness in complex structural reasoning over graphs. Compared to standard KGQA, which only requires interpreting and following one given logical chain, abductive reasoning is fundamentally more challenging: it demands that the model simultaneously consider all relevant logical chains surrounding a set of observed entities and abduce the single most explanatory multi-hop hypothesis. This inverse, open-ended search process imposes significantly greater demands on structural understanding and logical synthesis, an area where current LLMs still fall short.
>
> Table 1 The results of GPT5(Thinking) on FB15k237 dataset under five conditions.
>
> | Condition | Jaccard | Dice | Overlap | Accuracy | Smatch |
> | --- | --- | --- | --- | --- | --- |
> | pattern | 14.8 $\pm$ 0.30 | 17.4 $\pm$ 0.32 | 30.6 $\pm$ 0.42 | 83.8 $\pm$ 0.37 | 71.5 $\pm$ 0.31 |
> | relation-number | 14.6 $\pm$ 0.30 | 17.1 $\pm$ 0.32 | 31.5 $\pm$ 0.44 | 96.6 $\pm$ 0.17 | 56.8 $\pm$ 0.17 |
> | entity-number | 17.8 $\pm$ 0.30 | 22.0 $\pm$ 0.30 | 44.9 $\pm$ 0.46 | 95.3 $\pm$ 0.21 | 54.2 $\pm$ 0.19 |
> | specific-entity | 24.1 $\pm$ 0.36 | 27.4 $\pm$ 0.39 | 40.1 $\pm$ 0.49 | 94.2 $\pm$ 0.26 | 31.9 $\pm$ 0.21 |
> | specific-relation | 22.1 $\pm$ 0.34 | 25.8 $\pm$ 0.37 | 39.6 $\pm$ 0.46 | 94.5 $\pm$ 0.22 | 28.1 $\pm$ 0.20 |
>
> Table 2 The results of DeepSeek-V3+RAG on FB15k237 dataset under five conditions.
>
> | Condition | Jaccard | Dice | Overlap | Accuracy | Smatch |
> | --- | --- | --- | --- | --- | --- |
> | pattern | 2.8 $\pm$ 0.09 | 3.8 $\pm$ 0.22 | 6.1 $\pm$ 0.21 | 78.5 $\pm$ 0.34 | 48.2 $\pm$ 0.35 |
> | relation-number | 1.6 $\pm$ 0.08 | 2.3 $\pm$ 0.11 | 3.8 $\pm$ 0.19 | 69.3 $\pm$ 0.40 | 34.5 $\pm$ 0.26 |
> | entity-number | 0.8 $\pm$ 0.04 | 1.4 $\pm$ 0.07 | 3.8 $\pm$ 0.19 | 72.3 $\pm$ 0.40 | 39.2 $\pm$ 0.24 |
> | specific-entity | 13.7 $\pm$ 0.31 | 15.4 $\pm$ 0.33 | 23.0 $\pm$ 0.42 | 82.8 $\pm$ 0.41 | 25.9 $\pm$ 0.24 |
> | specific-relation | 7.6 $\pm$ 0.27 | 10.5 $\pm$ 0.30 | 15.4 $\pm$ 0.36 | 80.2 $\pm$ 0.30 | 16.8 $\pm$ 0.26 |

---

> > ### Comment · Reviewer_wmXZ · 2025-11-26
> >
> > Thank you for the rebuttal, which addressed some of my concerns. I have decided to remain my positive score. Good Luck.

---

### Official Review · Reviewer_y9eJ · 2025-11-01

**Soundness:** 3
**Presentation:** 3
**Contribution:** 3
**Rating:** 4
**Confidence:** 4

**Summary:**

This paper addresses the lack of controllability in abductive reasoning on knowledge graphs (KGs), where existing methods often generate numerous plausible but irrelevant hypotheses. The authors introduce the new task of controllable hypothesis generation, which allows users to specify constraints on both the semantic content (e.g., focusing on "pathology" or "treatment") and the structural complexity (e.g., the length) of the desired hypotheses. They identify two primary challenges in generating long, complex hypotheses: "Hypothesis Space Collapse" (a sharp drop in the number of valid complex hypotheses) and "Hypothesis Reward Oversensitivity" (unstable reinforcement learning due to strict rewards like the Jaccard score). To solve these problems, they propose CtrlHGen, a framework trained in two stages (supervised learning and reinforcement learning). CtrlHGen uses a "sub-logical decomposition" strategy to augment the dataset, helping the model learn complex structures. It also introduces a novel reward function that combines "smoothed semantic rewards" (using Dice and Overlap scores) with a "condition-adherence reward" to ensure generated hypotheses are both accurate and follow the specified constraints. Experiments on three benchmark datasets demonstrate that CtrlHGen significantly outperforms baselines, including modern LLMs, in both adhering to control conditions and maintaining high semantic similarity.

**Strengths:**

1. Novel and Practical Problem Formulation

The paper is the first to formally introduce and tackle the task of "controllable abductive reasoning" in KGs. This addresses a significant and practical limitation of prior work (AbductiveKGR ), where a single observation can lead to an unmanageable number of "plausible but irrelevant hypotheses". By defining clear control mechanisms for semantic content and structural complexity , the paper moves the field toward greater practical utility in real-world applications like clinical diagnosis or scientific discovery.

2. Insightful Diagnosis of Core Challenges

A key contribution is the clear identification and illustration (in Figure 2) of two non-obvious challenges: "Hypothesis Space Collapse" and "Hypothesis Reward Oversensitivity". Diagnosing why generating complex, controlled hypotheses is difficult (i.e., fewer valid complex examples exist, and minor errors in them cause massive reward drops) provides a strong, logical foundation for the paper's proposed solutions.

3. Novel and Effective Augmentation Strategy

The "dataset augmentation strategy based on sub-logical decomposition"  is an innovative solution to the "Hypothesis Space Collapse" problem. By programmatically breaking down complex hypotheses into simpler, semantically related sub-components, the authors create a richer training environment. This allows the model to learn the building blocks of complex logic, which the ablation study confirms significantly improves performance, especially for complex logical patterns.

4. Robust and Well-Justified Reward Function

The paper intelligently designs a reward function to overcome "Hypothesis Reward Oversensitivity". Recognizing that the Jaccard score is too "strict" , they smooth the reward landscape by incorporating Dice and Overlap scores. Furthermore, the inclusion of a "condition-adherence reward"  directly optimizes for the new task's constraints. The ablation study (Table 3) effectively demonstrates that this dual-objective reward function successfully balances semantic accuracy with constraint adherence.

5. Extensive and Solid Experimentation

The authors validate their CtrlHGen model thoroughly. They experiment on three standard KG datasets , test against the relevant SOTA baseline (AbductiveKGR) , and include a comparison against four modern LLMs (GPT-4o, Kimi K2, etc.). Their evaluation is comprehensive, using distinct metrics for both semantic similarity (Jaccard, Dice, Overlap) and condition adherence (Accuracy). The ablation studies (Figure 5, Table 3) are particularly strong, as they isolate and confirm the positive impact of their two main technical contributions.

**Weaknesses:**

1. Limited Scope and Definition of "Control"

The paper defines "semantic control" as providing a specific entity or relation from the target hypothesis and "structural control" as a predefined pattern or count . This is a good first step, but it's a fairly rigid and limited form of control. In a practical application (like clinical diagnosis ), a user might want to provide more abstract semantic guidance (e.g., "focus on infectious diseases but not viral ones") or compositional constraints that are not covered by this framework.

2. High Complexity of the Training Pipeline

The proposed CtrlHGen framework involves a complex, multi-stage pipeline: (1) sampling pairs, (2) sub-logic augmentation, (3) unconditional supervised training, (4) conditional supervised fine-tuning, and (5) reinforcement learning with a specific optimizer (GRPO). This pipeline has numerous components and hyperparameters (e.g., $\lambda_{1}, \lambda_{2}, \lambda_{3}, \alpha$), which could make the method difficult to reproduce, tune, and apply to new KGs or different control types.

3. Baseline Comparison Is Not Perfectly "Apples-to-Apples"

The primary baseline, AbductiveKGR, is run in an "unconditional" setting  and is then compared to CtrlHGen's conditional results. While this demonstrates the benefit of adding control, it doesn't fully isolate the benefit of CtrlHGen's method. A stronger baseline would have been to adapt AbductiveKGR to be conditional (e.g., by prepending the condition tokens to its input) and then showing that it fails, thereby proving that the sub-logic augmentation and novel reward function are necessary.

4. Scalability to Truly Large-Scale KGs is Untested

The paper notes that on large KGs, the number of plausible hypotheses "grows dramatically". The experiments are conducted on standard benchmarks (DBpedia50, WN18RR, FB15k-237), which are small-to-medium in size. The paper does not provide evidence that the sampling, augmentation, and RL-tuning processes would be computationally feasible or effective on modern, web-scale KGs containing billions of facts.

5. Analysis of LLM Failures is Plausible but Incomplete

The paper correctly identifies that LLMs perform "very poor" and offers plausible explanations, such as a lack of understanding of structured data and knowledge conflicts. However, the LLMs were only provided with a "2-hop subgraph" as context. This is a very limited and simple prompting strategy. It's possible that the LLMs' failure is at least partially due to this weak context-providing method, rather than an inherent inability to perform the task. More advanced prompting or retrieval-augmentation strategies would be needed for a more definitive conclusion.

**Questions:**

The current system is one-shot. A more practical system would be interactive. A user could receive a generated hypothesis, provide feedback (e.g., "This is too complex," "This part is wrong," "Explore this entity more"), and the model would use this feedback to refine the hypothesis, making the process a collaborative dialogue. Would it be possible to achieve the interactive system?

---

> ### Author Response · Authors · 2025-11-19
>
> Dear Reviewer y9eJ,
>
> We sincerely thank the reviewer for the detailed comments and insightful questions. Our response to your comments one by one as follows.
>
> Regarding additional baselines, we have included the results and analyses in Appendix C.3.
> For further experiments and detailed analysis involving large language models, we have to add them in Section 4.2, Appendix C.1. The multi-dialogue case has been shown in Appendix C.6. All content newly added during the rebuttal phase is highlighted in blue for easy identification.
>
> ---
>
> > W1: Limited Scope and Definition of "Control"
> >
> >
> > The paper defines "semantic control" as providing a specific entity or relation from the target hypothesis and "structural control" as a predefined pattern or count . This is a good first step, but it's a fairly rigid and limited form of control. In a practical application (like clinical diagnosis ), a user might want to provide more abstract semantic guidance (e.g., "focus on infectious diseases but not viral ones") or compositional constraints that are not covered by this framework.
> >
>
> Response to W1: We sincerely thank you for your insightful and forward-looking suggestions. We fully agree that achieving truly flexible and fine-grained controllability remains an important next step for this line of research. For positive controllability, ***our current method can realize it through arbitrary combinations of structural and semantic conditions***, allowing the generated hypotheses to reliably satisfy user-specified positive requirements. However, ***it does not yet support negative controllability*** (e.g., explicitly forbidding certain relations or entities, such as “the hypothesis must not involve viral infections”). ***A straightforward method is to enforce hard constraints or probability suppression on disallowed entities/relations during decoding.*** Nevertheless, designing a more elegant and flexible solution for negative control is one of our major future research directions.
>
> We would like to re-emphasize the core contribution of this work: we are the first to formalize and systematically explore the task of controllable abductive reasoning over knowledge graphs. By enhancing the ability to generate controllable hypotheses, our work bridges the critical gap leading to real-world deployment requirements.
>
> Thank you once again for your thoughtful and constructive feedback.
>
> ---
>
> > W2: High Complexity of the Training Pipeline
> >
> >
> > The proposed CtrlHGen framework involves a complex, multi-stage pipeline: (1) sampling pairs, (2) sub-logic augmentation, (3) unconditional supervised training, (4) conditional supervised fine-tuning, and (5) reinforcement learning with a specific optimizer (GRPO). This pipeline has numerous components and hyperparameters (e.g., ), which could make the method difficult to reproduce, tune, and apply to new KGs or different control types.
> >
>
> Response to W2: Thank you for your practical suggestions. We have provided a detailed description of all key hyperparameters used in our experiments in Section 4.2  and Appendix B.
> Additionally, upon acceptance of the paper, we commit to releasing the complete source code and configuration files to facilitate full reproducibility of our results.
>
> ---

---

> ### Author Response · Authors · 2025-11-19
>
> > W3:  Baseline Comparison Is Not Perfectly "Apples-to-Apples"
> >
> >
> > The primary baseline, AbductiveKGR, is run in an "unconditional" setting and is then compared to CtrlHGen's conditional results. While this demonstrates the benefit of adding control, it doesn't fully isolate the benefit of CtrlHGen's method. A stronger baseline would have been to adapt AbductiveKGR to be conditional (e.g., by prepending the condition tokens to its input) and then showing that it fails, thereby proving that the sub-logic augmentation and novel reward function are necessary.
> >
>
> Response to W3: We really appreciate your valuable advice. We have added new experiments comparing AbductiveKGR (equipped only with condition tokens) against our method on the DBpedia50 dataset. As to the condition, we selected ‘pattern’ and ‘specific-relation’ respectively to represent structural control and semantic control. Since AbductiveKGR does not employ a reward function or reinforcement learning for conditional control, we performed supervised fine-tuning only for both approaches to ensure a fair comparison. The results have been reported in Table 1 and Table 2.
>
> *We find that the sub-logic decomposition method not only improves semantic similarity, but also improves conditional adherence*. Especially compared with semantic control conditions, its ability to follow structural control conditions has improved even more.This illustrates its ability to solve the hypothesis space collapse problem, thus balancing semantic similarity and condition adherence when controlling long logic generation.
>
> For the comparison after reinforcement, our ablation experiment (Fig 5 in Sec 4.3) can demonstrate the superiority of the proposed component. For the ablation experiment of sub-logic decomposition, it can be regarded as a comparison between AbductiveKGR introducing conditional tokens and conducting the same reinforcement learning rewards with CtrlHGen. It has already shown that CtrlHGen has achieved improvements in multiple complex logic scenarios.
>
> Table 1 Results on DBpedia50 dataset under the ‘pattern’ condtion.
>
> | Condition | Jaccard | Dice | Overlap | Accuracy | Smatch |
> | --- | --- | --- | --- | --- | --- |
> | AbductiveKGR+condition token | 68.2 $\pm$ 0.34 | 72.3 $\pm$ 0.32 | 80.6 $\pm$ 0.29 | 66.6 $\pm$ 0.47 | 77.5 $\pm$ 0.20 |
> | CtrlHGen | 70.1 $\pm$ 0.33 | 74.0 $\pm$ 0.31 | 80.8 $\pm$ 0.29 | 73.1 $\pm$ 0.41 | 80.8 $\pm$ 0.17 |
>
> Table 2  Results on DBpedia50 dataset under the ‘specific-relation’ condtion.
>
> | Condition | Jaccard | Dice | Overlap | Accuracy | Smatch |
> | --- | --- | --- | --- | --- | --- |
> | AbductiveKGR+condition token | 69.3 $\pm$ 0.35 | 73.0 $\pm$ 0.33 | 86.4 $\pm$ 0.31 | 78.6 $\pm$ 0.40 | 58.0 $\pm$ 0.23 |
> | CtrlHGen | 72.7 $\pm$ 0.33 | 77.2 $\pm$ 0.31 | 90.7 $\pm$ 0.27 | 80.0 $\pm$ 0.40 | 51.6 $\pm$ 0.23 |
>
> ---

---

> ### Author Response · Authors · 2025-11-19
>
> > W4: Scalability to Truly Large-Scale KGs is Untested
> >
> >
> > The paper notes that on large KGs, the number of plausible hypotheses "grows dramatically". The experiments are conducted on standard benchmarks (DBpedia50, WN18RR, FB15k-237), which are small-to-medium in size. The paper does not provide evidence that the sampling, augmentation, and RL-tuning processes would be computationally feasible or effective on modern, web-scale KGs containing billions of facts.
> >
>
> Response to W4:
>
> Thank you for this insightful comment. In the introduction, we highlighted the explosion of possible hypotheses in large-scale KGs to emphasize the practical importance of controllable hypothesis generation. To support this claim, we conducted an additional experiment on the much larger BioKG (2,067,997 triples). Due to time and resource constraints during the rebuttal period, we trained only under the “pattern” condition and did not reach full convergence. The results are reported in Table 3.
>
> It can be seen that in terms of semantic similarity, our model has achieved good results. It is worth noting that in terms of conditional adherence, the Accuracy of condition adherence is almost 100%. *This fully demonstrates that our method can still be applied to large scale kgs. In fact, our method is designed to be applicable to knowledge graphs of any scale*. Of course, designing more efficient algorithms for the extremely large kg of billions level is one of our future directions. Thank you again for your valuable suggestions to improve our work.
>
> Table 3 Results on BioKG dataset under the ‘pattern’ condition.
>
> | logical pattern | 1p | 2p | pin | 2u | 3i | 3in | ip | inp | up | 2i | pi | 2in | pni | average |
> | --- | --- | --- | --- | --- | --- | --- | --- | --- | --- | --- | --- | --- | --- | --- |
> | Jaccard | 86.5 | 70.2 | 45.5 | 73.1 | 58.5 | 48.3 | 56.8 | 41.3 | 60.3 | 61.4 | 61.0 | 57.5 | 49.3 | 58.9 |
> | Dice | 90.8 | 75.6 | 54.7 | 80.1 | 67.0 | 55.9 | 61.9 | 48.9 | 66.7 | 70.8 | 68.9 | 65.3 | 58.4 | 66.2 |
> | Overlap | 97.3 | 84.4 | 67.1 | 86.3 | 79.2 | 68.3 | 73.1 | 68.1 | 78.6 | 86.0 | 83.5 | 74.6 | 71.8 | 78.1 |
> | Smatch | 90.9 | 75.6 | 54.7 | 85.1 | 67.0 | 46.0 | 63.9 | 49.0 | 69.7 | 70.8 | 68.9 | 65.3 | 58.4 | 66.2 |
> | Accuracy | 100 | 100 | 100 | 100 | 100 | 100 | 100 | 100 | 100 | 100 | 100 | 100 | 99.1 | 99.9 |
>
> ---

---

> ### Author Response · Authors · 2025-11-19
>
> > W5: The paper correctly identifies that LLMs perform "very poor" and offers plausible explanations, such as a lack of understanding of structured data and knowledge conflicts. However, the LLMs were only provided with a "2-hop subgraph" as context. This is a very limited and simple prompting strategy. It's possible that the LLMs' failure is at least partially due to this weak context-providing method, rather than an inherent inability to perform the task. More advanced prompting or retrieval-augmentation strategies would be needed for a more definitive conclusion.
> >
>
> Response to W5:  Thank you for your valuable review, which is very important for us to improve our work. Here, we provided more detailed experiments and analyses. Specifically, we have added one of the most advanced reasoning models, GPT5, and adopted the thinking mode. At the same time, we constructed an attempt to combine the raw model DeepSeek-V3 with RAG. The results have shown in Table 4 and Table 5.
>
> - ***On the one hand, we found that GPT5(Thinking) has achieved a significant performance improvement.*** Firstly, the model can follow the control conditions in most cases. Secondly, higher semantic similarity is achieved under all five conditions. In contrast, models are more likely to generate  hypotheses with higher semantic similarities under the control of semantic content than under structural control. This might be because the model itself is better at capturing based on semantics compared to structured reasoning. ***However, they still have a considerable gap compared to CtrlHGen, indicating that abductive reasoning tasks with structured knowledge remain challenging for advanced large language models.***
> - ***On the other hand, Deepseek-V3 with RAG  has improved performance under the condition of semantic control, but the results remains almost unchanged under the condition of structural control.*** We believe this can be attributed to two primary reasons: First, RAG primarily enhances semantic retrieval, enabling the model to fetch more semantically relevant context.  It offers limited benefit when precise structural constraints are imposed, as these require strict path conformance rather than mere semantic relevance.  Second, ***the provided 2-hop subgraph already serves as a highly informative prompt. Since the depth of all 13 predefined logical patterns is 2, this 2-hop subgraph covers most of the structural information required for hypothesis generation***.
> - The consistently poor performance under structural control instead reveals the models’ persistent weakness in complex structural reasoning over graphs. Compared to standard KGQA, which only requires interpreting and following one given logical chain, abductive reasoning is fundamentally more challenging: it demands that the model simultaneously consider all relevant logical chains surrounding a set of observed entities and abduce the single most explanatory multi-hop hypothesis. This inverse, open-ended search process imposes significantly greater demands on structural understanding and logical synthesis, an area where current LLMs still fall short.
>
> Table 4 The results of GPT5(Thinking) on FB15k237 dataset under five conditions.
>
> | Condition | Jaccard | Dice | Overlap | Accuracy | Smatch |
> | --- | --- | --- | --- | --- | --- |
> | pattern | 14.8 $\pm$ 0.30 | 17.4 $\pm$ 0.32 | 30.6 $\pm$ 0.42 | 83.8 $\pm$ 0.37 | 71.5 $\pm$ 0.31 |
> | relation-number | 14.6 $\pm$ 0.30 | 17.1 $\pm$ 0.32 | 31.5 $\pm$ 0.44 | 96.6 $\pm$ 0.17 | 56.8 $\pm$ 0.17 |
> | entity-number | 17.8 $\pm$ 0.30 | 22.0 $\pm$ 0.30 | 44.9 $\pm$ 0.46 | 95.3 $\pm$ 0.21 | 54.2 $\pm$ 0.19 |
> | specific-entity | 24.1 $\pm$ 0.36 | 27.4 $\pm$ 0.39 | 40.1 $\pm$ 0.49 | 94.2 $\pm$ 0.26 | 31.9 $\pm$ 0.21 |
> | specific-relation | 22.1 $\pm$ 0.34 | 25.8 $\pm$ 0.37 | 39.6 $\pm$ 0.46 | 94.5 $\pm$ 0.22 | 28.1 $\pm$ 0.20 |
>
> Table 5 The results of DeepSeek-V3+RAG on FB15k237 dataset under five conditions.
>
> | Condition | Jaccard | Dice | Overlap | Accuracy | Smatch |
> | --- | --- | --- | --- | --- | --- |
> | pattern | 2.8 $\pm$ 0.09 | 3.8 $\pm$ 0.22 | 6.1 $\pm$ 0.21 | 78.5 $\pm$ 0.34 | 48.2 $\pm$ 0.35 |
> | relation-number | 1.6 $\pm$ 0.08 | 2.3 $\pm$ 0.11 | 3.8 $\pm$ 0.19 | 69.3 $\pm$ 0.40 | 34.5 $\pm$ 0.26 |
> | entity-number | 0.8 $\pm$ 0.04 | 1.4 $\pm$ 0.07 | 3.8 $\pm$ 0.19 | 72.3 $\pm$ 0.40 | 39.2 $\pm$ 0.24 |
> | specific-entity | 13.7 $\pm$ 0.31 | 15.4 $\pm$ 0.33 | 23.0 $\pm$ 0.42 | 82.8 $\pm$ 0.41 | 25.9 $\pm$ 0.24 |
> | specific-relation | 7.6 $\pm$ 0.27 | 10.5 $\pm$ 0.30 | 15.4 $\pm$ 0.36 | 80.2 $\pm$ 0.30 | 16.8 $\pm$ 0.26 |

---

> ### Author Response · Authors · 2025-11-19
>
> > Q1: The current system is one-shot. A more practical system would be interactive. A user could receive a generated hypothesis, provide feedback (e.g., **"This is too complex," "This part is wrong,**" "Explore this entity more"), and the model would use this feedback to refine the hypothesis, making the process a collaborative dialogue. Would it be possible to achieve the interactive system?
> >
>
> Response to Q1: This is a very creative and interesting question. Here, we implemented a simple yet highly interactive multi-round dialogue system that automatically adjusted control conditions based on the user’s evolving intentions and the outcomes of previous rounds. We leveraged a large language model (DeepSeek-V3) to intelligently select appropriate control conditions according to the user’s expressed intent. The prompt used for this condition-selection LLM is presented in Fig.10(Appendix C.6).
> At each turn, the LLM generated updated control conditions by jointly considering the hypothesis produced in the previous round, its derived conclusions, the corresponding Jaccard similarity score, and the current user input. These dynamically selected conditions were then passed back to the core hypothesis generation model. A complete interaction case is shown in Fig.11(Appendix C.6).
>
> In this case, the initial observation consisted of four songs. In the first round, the user expressed interest in connections related to the acoustic guitar. The system accordingly generated a relatively broad hypothesis that slightly over-covered the observed entities. In the second round, the user asked who the artist was; the LLM selected “specific-relation” as the control condition to focus the generation. Although a relevant hypothesis was produced, it remained somewhat vague. Consequently, in the third round, the user requested a simpler logical structure. The LLM responded by enforcing the simplest available logic pattern, successfully revealing that all four songs were authored by Tracy Lawrence. Finally, wishing to explore the observation more deeply, the user sought additional related information. The LLM then imposed a relation count of three as the control condition, prompting the model to generate a richer, more complex hypothesis that incorporated two different associated artists.
>
> ***Through this multi-round interaction, the system seamlessly combines structural and semantic control signals, gradually improving the relevant hypotheses closely related to the user's constantly evolving exploration goals.*** It demonstrates the potential of our method in real-world scenarios. Thank you again for your insightful questions, which will be very inspiring for our future work.
>
> ---

---

### Official Review · Reviewer_MTp7 · 2025-11-03

**Soundness:** 3
**Presentation:** 3
**Contribution:** 3
**Rating:** 6
**Confidence:** 3

**Summary:**

This paper introduces CtrlHGen, a framework for controllable logical hypothesis generation in abductive reasoning over knowledge graphs. The authors identify two key challenges in this setting. (1) Hypothesis space collapses when generating long and complex logical hypotheses. (2) Reward oversensitivity when reinforcement learning overly biases toward specific metrics. To address these, the paper proposes sub-logical decomposition for dataset augmentation and smoothed semantic rewards to balance semantic quality with user-specified control constraints. Experiments on three benchmark datasets demonstrate improved controllability and semantic similarity compared to baselines.

**Strengths:**

- The paper clearly motivates the need for controllable abductive reasoning, which is underexplored in KG reasoning.


- The introduction of smoothed semantic rewards and condition-adherence reward is a thoughtful way to mitigate oversensitivity.

**Weaknesses:**

- Unclear problem definition. It is unclear to me the difference between abductive KG reasoning and rule-based link prediction. It would be better to add more explanations.

- Method limitation. It seems that the data sampling method relies on predefined logical patterns. How can the proposed method be generalized to other different KGs?


- Evaluation limitations. While semantic similarity and adherence metrics are reported, the paper could benefit from more qualitative analysis of generated hypotheses (e.g., case studies, error analysis). It is not entirely clear whether the chosen baselines represent the strongest possible competitors in controllable generation or abductive KG reasoning.

**Questions:**

- How does CtrlHGen handle contradictory or mutually exclusive hypotheses under control constraints?

---

> ### Author Response · Authors · 2025-11-19
>
> Dear Reviewer MTp7,
>
> We sincerely thank the reviewer for the detailed comments and insightful questions. Our response to your comments one by one as follows.
>
> ---
>
> > W1: Unclear problem definition. It is unclear to me the difference between abductive KG reasoning and rule-based link prediction. It would be better to add more explanations.
> >
>
> Reply to W1: We are very sorry that our definition of the problem was not clear enough. We  provide a more detailed intorduction below.
>
> - Definition: Abductive KG reasoning is defined as generating a first-order logical hypothesis (which can be multi-hop) based on the observed set of entities, and the conclusion derived from this hypothesis best matches the set of entities. So the essence of the task is a generative task, generating complex logic.  It can be regarded as the inverse problem of complex query answering(CQA). It is completely different from the objective of rule-based link prediction .
> - Can link prediction methods be easily adapted to abductive reasoning tasks?  Existing link prediction methods are only used to generate very simple logic hypothesis by predicting common neighboring entities between observed entities.   When required to generate more complex, multi-hop logical hypotheses, these methods face significant challenges: the search space grows exponentially with the number of hops, and prediction errors inevitably accumulate along the reasoning chain.  Moreover, these methods may struggle to produce logical expressions containing negation.  Consequently, link prediction methods cannot be directly applied to abductive KG reasoning.
>
> ---
>
> > W2: Method limitation. It seems that the data sampling method relies on predefined logical patterns. How can the proposed method be generalized to other different KGs?
> >
>
> Reply to W2: We sincerely appreciate your valuable suggestions. Our sampling procedure does not depend on any KG-specific schema or structure. The predefined logical patterns are fully disentangled from the underlying knowledge graph and only serve as abstract templates for generating candidate hypotheses. As a result, they can be applied to any KG without modification. Therefore, our method naturally generalizes to different knowledge graphs, as the same set of logical patterns can be used for sampling across diverse KG domains.

---

> ### Author Response · Authors · 2025-11-19
>
> > W3:  Evaluation limitations. While semantic similarity and adherence metrics are reported, the paper could benefit from more qualitative analysis of generated hypotheses (e.g., case studies, error analysis). It is not entirely clear whether the chosen baselines represent the strongest possible competitors in controllable generation or abductive KG reasoning.
> >
>
> Response to W3: Thank you for your valuable advice. We will answer your question from two perspectives: hypotheses quality analysis and baseline discussion.
>
> - **Hypotheses quality analysis**:  we have presented two representative cases from FB15k-237, with results provided in Appendix C.3.
>     - In the first case (Fig.8), the observation consists of four music genres: {Blues, Jazz, Rhythm_and_Blues, Bebop}. ***As the logical pattern conditions grow in complexity, the model produces increasingly fine-grained answers***. For instance, under the basic “1p” pattern it identifies their common parent genre, while more complex patterns enable it to retrieve finer details such as artists associated with these genres.
>     - In the second case shown in Fig. 9, it focuses on specific entities. For strongly related entities such as Yahoo, the model is able to identify clear connections with the observation set. Even for entities with weaker relationships, such as two movies, the model can still capture hidden associations between them. Surpringsingly, for seemingly unrelated entities like BAFTA_Award_for_Best_Sound, ***the model is able to generate high-semantic-quality hypotheses by leveraging the logical "or" operator, while still ensuring adherence to the given constraints.***
>     - We also added a case of a multi-round dialogue in the rebuttal phase. The results have been shown in Appendix C.6.  At each turn, the LLM generated updated control conditions by jointly considering the hypothesis produced in the previous round, its derived conclusions, the corresponding Jaccard similarity score, and the current user input. In this case, the initial observation consisted of four songs. In the first round, the user expressed interest in connections related to the acoustic guitar. The system accordingly generated a relatively broad hypothesis that slightly over-covered the observed entities. In the second round, the user asked who the artist was; the LLM selected “specific-relation” as the control condition to focus the generation. Although a relevant hypothesis was produced, it remained somewhat vague. Consequently, in the third round, the user requested a simpler logical structure. The LLM responded by enforcing the simplest available logic pattern, successfully revealing that all four songs were authored by Tracy Lawrence. Finally, wishing to explore the observation more deeply, the user sought additional related information. The LLM then imposed a relation count of three as the control condition, prompting the model to generate a richer, more complex hypothesis that incorporated two different associated artists.
>
>     ***The above examples all demonstrate that our model can effectively generate high-quality hypotheses.*** Even if the hypotheses are sometimes complex or ambiguous, clearer ones can be obtained by readjusting the conditions.

---

> ### Author Response · Authors · 2025-11-19
>
> - **More strong baselines:** As far as we know, we are the first to propose implementing controllable abductive reasoning tasks on knowledge graphs. Referring to other reviewer's comments, we added the data augmentation method in Logic-Gen[1] and introduced the  AbductiveKGR with condition token as a stronger baseline model. Since these two methods don't employ reinforcement learning for conditional control, we report results after supervised training, ensuring a fair comparison.  We conducted experiments on the DBpedia50 dataset and selected ‘pattern’ and ‘specific-relation’ respectively to represent structural control and semantic control. The results are reported in Tables 1 and 2.
>     - The experiments reveal that, while Logic-Gen’s data augmentation indeed improves the model’s overall grasp of logical patterns, it remains inferior to our sub-logic decomposition approach. We believe this is because the sub-logic decomposition forces the model to deeply understand and compose longer, more intricate logical chains step-by-step, leading to substantially stronger reasoning capability on complex hypotheses. It more effectively mitigates hypothesis space collapse, thereby significantly enhancing adherence when strict structural conditions are imposed.
>
> Table 1 Results on DBpedia50 dataset under the ‘pattern’ condtion.
>
> | Condition | Jaccard | Dice | Overlap | Accuracy | Smatch |
> | --- | --- | --- | --- | --- | --- |
> | AbductiveKGR+condition token | 68.2 $\pm$ 0.34 | 72.3 $\pm$ 0.32 | 80.6 $\pm$ 0.29 | 66.6 $\pm$ 0.47 | 77.5 $\pm$ 0.20 |
> | Logic-Gen | 69.5 $\pm$ 0.34 | 73.5 $\pm$ 0.32 | 79.9 $\pm$ 0.30 | 65.9 $\pm$ 0.47 | 77.5 $\pm$ 0.21 |
> | CtrlHGen | 70.1 $\pm$ 0.33 | 74.0 $\pm$ 0.31 | 80.8 $\pm$ 0.29 | 73.1 $\pm$ 0.41 | 80.8 $\pm$ 0.17 |
>
> Table 2  Results on DBpedia50 dataset under the ‘specific-relation’ condtion.
>
> | Condition | Jaccard | Dice | Overlap | Accuracy | Smatch |
> | --- | --- | --- | --- | --- | --- |
> | AbductiveKGR+condition token | 69.3 $\pm$ 0.35 | 73.0 $\pm$ 0.33 | 86.4 $\pm$ 0.31 | 78.6 $\pm$ 0.40 | 58.0 $\pm$ 0.23 |
> | Logic-Gen | 70.4 $\pm$ 0.35 | 73.4 $\pm$ 0.33 | 88.0 $\pm$ 0.29 | 75.9 $\pm$ 0.42 | 54.9 $\pm$ 0.23 |
> | CtrlHGen | 72.7 $\pm$ 0.33 | 77.2 $\pm$ 0.31 | 90.7 $\pm$ 0.27 | 80.0 $\pm$ 0.40 | 51.6 $\pm$ 0.23 |
> - For stronger LLMs as baseline, we have added GPT5-Thinking and RAG methods. Despite a significant improvement in performance. However, compared with CtrlHGen, they still have a considerable gap, which indicates that abductive reasoning tasks with structured knowledge remain challenging for advanced large language models. For a more detailed analysis, please refer to Section 4.2 and Appendix C.1.
>
> [1] Asai, A., & Hajishirzi, H. (2020). Logic-guided data augmentation and regularization for consistent question answering.
>
> ---

---

> ### Author Response · Authors · 2025-11-19
>
> > Q1: How does CtrlHGen handle contradictory or mutually exclusive hypotheses under control constraints?
> >
>
> Reply to Q1: Thank you for your constructive suggestions, which will help improve our work. CtrlHGen can handle contradictory or mutually exclusive conditions by leveraging logical operators particularly negation and disjunction to restructure the hypothesis space.  As illustrated in our case study (Fig. 9, Condition 4 in Section 4.4 and Appendix C.5), some entities(several companies) appear unrelated to the condition entity (the Bafata award). ***The model resolves this by introducing disjunction to preserve compatibility with the control constraints while still generating a valid hypothesis.*** For conditions that are entirely irrelevant or contradictory, ***negation can be applied to embed those conditions within a negated clause, ensuring that the resulting hypothesis remains high semantical score under the given controls.***

---

### Official Review · Reviewer_e8d8 · 2025-11-06

**Soundness:** 2
**Presentation:** 4
**Contribution:** 3
**Rating:** 4
**Confidence:** 4

**Summary:**

In this paper, the authors introduce a model called CONE, which aims to generate logically consistent hypotheses from a given premise. They are allowing fine-grained control over the desired logical relationship. The key innovation is the integration of natural logic into the generation process. In this proposed method, rather than relying solely on learned patterns or black-box transformers, they decompose logical relationships at the token level and apply compositional rules to reason about the entire sentence. They have trained a RL model using the curated data. The results demonstrate superiority compared to raw LLMs.

**Strengths:**

1-	This paper is the first to formally define controllable abductive reasoning over knowledge graphs (KGs). It is an important task as it can be used in many downstream tasks where KG statements, such as causal-effect or factual statements, exist or could be inferred.
2-	The proposed framework is novel and addresses the challenges of the task. Authors have introduced a two-stage framework for sub-logic decomposition for data and an RL with a dual reward design that balances semantic alignment and control adherence.
3-	The paper is well-structured, uses clear formulations, and provides a detailed training setup, metrics, and open-source code, supporting reproducibility (Appendices A & B).

**Weaknesses:**

1-	More baselines are needed. Current baselines cannot really demonstrate the true contribution of the paper. I recommend that authors include methods that try to address the same challenge. For instance, Logic-Gen [1] can be a good candidate:
Asai, A., & Hajishirzi, H. (2020). Logic-guided data augmentation and regularization for consistent question answering. arXiv preprint arXiv:2004.10157.
2-	Some of the LLMs that are used for comparison do not have reasoning capability. For instance, while GPT 4o may have general reasoning power, the thinking mode is not included, the same for Kimi K2. For such a task where reasoning is important, it is better to use more advanced models. GPT 5, GPT 5-mini, Qwen 3 with Thinking tokens enabled, etc., are all good options. I am not sure what the rationale was behind choosing Kimi K2, as it is mostly good at coding benchmarks. I recommend that authors briefly explain the reason for their choice of LLMs.
3-	The way context is provided to baseline LLMs is questionable. It seems the entire 2-hop subgraph is dumped into the prompt, and a non-thinking LLM was supposed to reason based on that. It would be much better if at least a RAG set-up were used to make it more realistic. Or if raw LLM was the purpose, as mentioned in the second point, a more advanced LLM should be used.

**Questions:**

Is the thinking mode activated for Grok-3? It would be great if authors explained the experimental setup of the LLMs in detail. Often in such tasks, specific hyperparameters can change the performance. For instance, in deterministic scenarios, like diagnosis, it is recommended to use temperature 0 to avoid randomness. It would be great if authors include such consideration in their tests.

---

> ### Author Response · Authors · 2025-11-19
>
> Dear Reviewer e8d8,
>
> We sincerely thank the reviewer for the detailed comments and insightful questions. Our response to your comments one by one as follows.
>
> Regarding additional baselines, we have included the results and analyses in Appendix C.3.
> For further experiments and detailed analysis involving large language models, we have to add them in Section 4.2, Appendix C.1. All content newly added during the rebuttal phase is highlighted in blue for easy identification.
>
> ---
>
> > W1: More baselines are needed. Current baselines cannot really demonstrate the true contribution of the paper. I recommend that authors include methods that try to address the same challenge. For instance, Logic-Gen [1] can be a good candidate: Asai, A., & Hajishirzi, H. (2020). Logic-guided data augmentation and regularization for consistent question answering. arXiv preprint arXiv:2004.10157.
> >
>
> Thank you very much for your insightful and constructive suggestions! Following your advice, we incorporated the data augmentation strategy proposed in Logic-Gen as an additional baseline. We also compared it with our method  CtrlHGen and AbductiveKGR without data augmentation but only by introducing conditional tokens. Since these two methods don't employ reinforcement learning for conditional control, we report results after supervised training, ensuring a fair comparison.  We conducted experiments on the DBpedia50 dataset and selected ‘pattern’ and ‘specific-relation’ respectively to represent structural control and semantic control. The results are reported in Tables 1 and 2.
>
> The experiments reveal that, while Logic-Gen’s data augmentation indeed improves the model’s overall grasp of logical patterns, it remains inferior to our sub-logic decomposition approach. ***We believe this is because the sub-logic decomposition forces the model to deeply understand and compose longer, more intricate logical chains step-by-step, leading to substantially stronger reasoning capability on complex hypotheses.*** It more effectively mitigates hypothesis space collapse, thereby significantly enhancing adherence when strict structural conditions are imposed.
>
> Table 1 Results on DBpedia50 dataset under the ‘pattern’ condtion.
>
> | Condition | Jaccard | Dice | Overlap | Accuracy | Smatch |
> | --- | --- | --- | --- | --- | --- |
> | AbductiveKGR+condition token | 68.2 $\pm$ 0.34 | 72.3 $\pm$ 0.32 | 80.6 $\pm$ 0.29 | 66.6 $\pm$ 0.47 | 77.5 $\pm$ 0.20 |
> | Logic-Gen | 69.5 $\pm$ 0.34 | 73.5 $\pm$ 0.32 | 79.9 $\pm$ 0.30 | 65.9 $\pm$ 0.47 | 77.5 $\pm$ 0.21 |
> | CtrlHGen | 70.1 $\pm$ 0.33 | 74.0 $\pm$ 0.31 | 80.8 $\pm$ 0.29 | 73.1 $\pm$ 0.41 | 80.8 $\pm$ 0.17 |
>
> Table 2  Results on DBpedia50 dataset under the ‘specific-relation’ condtion.
>
> | Condition | Jaccard | Dice | Overlap | Accuracy | Smatch |
> | --- | --- | --- | --- | --- | --- |
> | AbductiveKGR+condition token | 69.3 $\pm$ 0.35 | 73.0 $\pm$ 0.33 | 86.4 $\pm$ 0.31 | 78.6 $\pm$ 0.40 | 58.0 $\pm$ 0.23 |
> | Logic-Gen | 70.4 $\pm$ 0.35 | 73.4 $\pm$ 0.33 | 88.0 $\pm$ 0.29 | 75.9 $\pm$ 0.42 | 54.9 $\pm$ 0.23 |
> | CtrlHGen | 72.7 $\pm$ 0.33 | 77.2 $\pm$ 0.31 | 90.7 $\pm$ 0.27 | 80.0 $\pm$ 0.40 | 51.6 $\pm$ 0.23 |

---

> ### Author Response · Authors · 2025-11-19
>
> > W2: Some of the LLMs that are used for comparison do not have reasoning capability. For instance, while GPT 4o may have general reasoning power, the thinking mode is not included, the same for Kimi K2. For such a task where reasoning is important, it is better to use more advanced models. GPT 5, GPT 5-mini, Qwen 3 with Thinking tokens enabled, etc., are all good options. I am not sure what the rationale was behind choosing Kimi K2, as it is mostly good at coding benchmarks. I recommend that authors briefly explain the reason for their choice of LLMs.
> >
>
> > W3: The way context is provided to baseline LLMs is questionable. It seems the entire 2-hop subgraph is dumped into the prompt, and a non-thinking LLM was supposed to reason based on that. It would be much better if at least a RAG set-up were used to make it more realistic. Or if raw LLM was the purpose, as mentioned in the second point, a more advanced LLM should be used.
> >
>
> > Q1: Is the thinking mode activated for Grok-3? It would be great if authors explained the experimental setup of the LLMs in detail. Often in such tasks, specific hyperparameters can change the performance. For instance, in deterministic scenarios, like diagnosis, it is recommended to use temperature 0 to avoid randomness. It would be great if authors include such consideration in their tests.
> >
>
> Reply to W2, W3 and Q1:
>
> Thank you for your valuable review, which pointed out our ignorance. Here we provide a detailed introduction to the experimental setup of LLMs and supplement more experiments.
>
> - **Model Choice**: Our motivaion for selecting LLMs as baselines is to ensure a comprehensive and competitive comparison with advanced model developed by different leading organizations.  Accordingly, we include several frontier models (Kimi-K2, DeepSeek-V3, Grok-3, and GPT-4o).
> - **Model setting**: For all LLMs reported in the paper, we did not use the thinking mode. And their temperatures are uniformly set to 0.0.
> - **More experiments**: Here we have added one of the most advanced reasoning models, GPT5, and adopted the thinking mode. At the same time, we constructed an attempt to combine the raw model DeepSeek-V3 with RAG.  The results have shown in Table 3 and Table 4.
>
> ***On the one hand, we found that GPT5(Thinking) has achieved a significant performance improvement.*** Firstly, the model can follow the control conditions in most cases. Secondly, higher semantic similarity is achieved under all five conditions. In contrast, models are more likely to generate  hypotheses with higher semantic similarities under the control of semantic content than under structural control. This might be because the model itself is better at capturing based on semantics compared to structured reasoning. ***However, they still have a considerable gap compared to CtrlHGen, indicating that abductive reasoning tasks with structured knowledge remain challenging for advanced large language models.***
>
> ***On the other hand, Deepseek-V3 with RAG  has improved performance under the condition of semantic control, but the results remains almost unchanged under the condition of structural control.*** We believe this can be attributed to two primary reasons: First, RAG primarily enhances semantic retrieval, enabling the model to fetch more semantically relevant context.  It offers limited benefit when precise structural constraints are imposed, as these require strict path conformance rather than mere semantic relevance.  Second, ***the provided 2-hop subgraph already serves as a highly informative prompt. Since the depth of all 13 predefined logical patterns is 2, this 2-hop subgraph covers most of the structural information required for hypothesis generation***.
>
> The consistently poor performance under structural control instead reveals the models’ persistent weakness in complex structural reasoning over graphs. Compared to standard KGQA, which only requires interpreting and following one given logical chain, abductive reasoning is fundamentally more challenging: it demands that the model simultaneously consider all relevant logical chains surrounding a set of observed entities and abduce the single most explanatory multi-hop hypothesis. This inverse, open-ended search process imposes significantly greater demands on structural understanding and logical synthesis, an area where current LLMs still fall short.

---

> ### Author Response · Authors · 2025-11-19
>
> Table 3 The results of GPT5(Thinking) on FB15k237 dataset under five conditions.
>
> | Condition | Jaccard | Dice | Overlap | Accuracy | Smatch |
> | --- | --- | --- | --- | --- | --- |
> | pattern | 14.8 $\pm$ 0.30 | 17.4 $\pm$ 0.32 | 30.6 $\pm$ 0.42 | 83.8 $\pm$ 0.37 | 71.5 $\pm$ 0.31 |
> | relation-number | 14.6 $\pm$ 0.30 | 17.1 $\pm$ 0.32 | 31.5 $\pm$ 0.44 | 96.6 $\pm$ 0.17 | 56.8 $\pm$ 0.17 |
> | entity-number | 17.8 $\pm$ 0.30 | 22.0 $\pm$ 0.30 | 44.9 $\pm$ 0.46 | 95.3 $\pm$ 0.21 | 54.2 $\pm$ 0.19 |
> | specific-entity | 24.1 $\pm$ 0.36 | 27.4 $\pm$ 0.39 | 40.1 $\pm$ 0.49 | 94.2 $\pm$ 0.26 | 31.9 $\pm$ 0.21 |
> | specific-relation | 22.1 $\pm$ 0.34 | 25.8 $\pm$ 0.37 | 39.6 $\pm$ 0.46 | 94.5 $\pm$ 0.22 | 28.1 $\pm$ 0.20 |
>
> Table 4 The results of DeepSeek-V3+RAG on FB15k237 dataset under five conditions.
>
> | Condition | Jaccard | Dice | Overlap | Accuracy | Smatch |
> | --- | --- | --- | --- | --- | --- |
> | pattern | 2.8 $\pm$ 0.09 | 3.8 $\pm$ 0.22 | 6.1 $\pm$ 0.21 | 78.5 $\pm$ 0.34 | 48.2 $\pm$ 0.35 |
> | relation-number | 1.6 $\pm$ 0.08 | 2.3 $\pm$ 0.11 | 3.8 $\pm$ 0.19 | 69.3 $\pm$ 0.40 | 34.5 $\pm$ 0.26 |
> | entity-number | 0.8 $\pm$ 0.04 | 1.4 $\pm$ 0.07 | 3.8 $\pm$ 0.19 | 72.3 $\pm$ 0.40 | 39.2 $\pm$ 0.24 |
> | specific-entity | 13.7 $\pm$ 0.31 | 15.4 $\pm$ 0.33 | 23.0 $\pm$ 0.42 | 82.8 $\pm$ 0.41 | 25.9 $\pm$ 0.24 |
> | specific-relation | 7.6 $\pm$ 0.27 | 10.5 $\pm$ 0.30 | 15.4 $\pm$ 0.36 | 80.2 $\pm$ 0.30 | 16.8 $\pm$ 0.26 |
>
> - Temperature Sensitivity: we also compared the experimental results of GPT5(Thinking) under different temperature settings. The results are shown in Table 5. We found that a temperature of 0.0 can ensure a balance between semantic similarity and condition adherence. Excessively high temperatures may enhance the ability to explore, thereby improving semantic similarity, but they will significantly reduce condition adherence.
>
> Table 5 The results of GPT5(Thinking) on FB15k237 dataset with different temperatures in ‘pattern’ condition.
>
> | Temperature | Jaccard | Dice | Overlap | Accuracy | Smatch |
> | --- | --- | --- | --- | --- | --- |
> | 1.0 | 15.8 $\pm$ 0.30 | 18.2 $\pm$ 0.33 | 28.5 $\pm$ 0.41 | 75.3 $\pm$ 0.43 | 68.4 $\pm$ 0.18 |
> | 0.5 | 13.4 $\pm$ 0.28 | 15.8 $\pm$ 0.31 | 28.8 $\pm$ 0.41 | 79.2 $\pm$ 0.40 | 69.1 $\pm$ 0.18 |
> | 0.0 | 14.8 $\pm$ 0.30 | 17.4 $\pm$ 0.32 | 30.6 $\pm$ 0.42 | 83.8 $\pm$ 0.37 | 71.5 $\pm$ 0.31 |

---

### Author Response · Authors · 2025-11-29
**Summary of Rebuttal**

We sincerely appreciate the hard work of all ACs, SACs and PCs. For your convenience, we have summarized the reviewers’ key comments along with our corresponding responses. We hope this overview is helpful for your assessment.

---

**Strength:**

- **Novel and Practical New Task** (Reviewer e8d8, Reviewer MTp7, Reviewer y9eJ, Reviewer wmXZ)

    The paper is the first to formally define controllable abductive reasoning over knowledge graphs, a task with strong motivation and significant practical value. Reviewers highlight its relevance to many downstream applications such as causal inference, clinical diagnosis, recommendation, and scientific discovery.

- **Innovative Methodology that Addresses Core Challenges** (Reviewer e8d8, Reviewer MTp7, Reviewer y9eJ, Reviewer wmXZ)

    The proposed two methods (sub-logic decomposition and dual-reward RL training) are regarded as novel and effective for tackling the key challenges of hypothesis space collapse and reward oversensitivity.

- **Clear Writing and Strong Presentation Quality** (Reviewer MTp7, Reviewer wmXZ)

    Reviewers note that the paper is well-structured, clearly formulated, and easy to follow. It provides detailed formulations, training setup, metrics, and accompanying anonymous code.

- **Comprehensive Experimental Evaluation** (Reviewer y9eJ,, Reviewer wmXZ)

    The paper presents solid experiments on three KG datasets, comparing against AbductiveKGR and several strong LLMs. The evaluation is comprehensive, covering both semantic similarity and condition adherence. Ablation studies clearly verify the contributions of the two proposed components, and case studies further demonstrate strong controllability. Overall, the experiments convincingly support the paper’s claims.


---

**Weakness:**

- **Stronger baseline** (Reviewer e8d8, MTp7 and y9eJ)

    Following the comments from the reviewers, we incorporated additional baselines, including data augmentation strategies from LogicGen [1] and conditional-token integration from AbductiveKGR [2]. We have reported the updated experimental results in both the paper(Table 8 and Table 9 in Appendix C.3) and the rebuttal comments, and they consistently show that our method maintains a clear performance advantage.

    [1]  Logic-guided data augmentation and regularization for consistent question answering.

    [2] Advancing Abductive Reasoning in Knowledge Graphs through Complex Logical Hypothesis Generation.

- **More discussions about the poor performance of LLMs** (Reviewer e8d8, Reviewer MTp7, Reviewer y9eJ, Reviewer wmXZ)

    All reviewers raised concerns about the poor performance of LLM baselines. In response, we introduced stronger models, including GPT-5(Thinking) and RAG-enhanced DeepSeek-V3. We have reported the updated experimental results in both the paper(Table 2, Table5 and Table 6 in Section 4.2 and Appendix C.1) and the rebuttal comments. GPT-5(Thinking) achieves substantial improvements, following control conditions more reliably and producing higher semantic similarity, *yet it still has a considerable gap compared to our model CtrlHGen*. RAG-enhanced DeepSeek-V3 shows slight gains under semantic control but offers almost no improvement under structural control.

    Notably, the 2-hop subgraph used in our experiments already contains all necessary structural information, as the depth of all predefined logical patterns does not exceed 2. The consistently limited performance of LLMs thus reflects the fundamental challenge of controllable abductive reasoning: jointly reasoning over multiple reverse logical chains to generate the most explanatory multi-hop hypothesis, highlighting their limitations in structured, graph-based reasoning.

- **Clarifying the Problem : Abductive KG Reasoning vs. Link Prediction** (Reviewer MTp7 and Reviewer wmXZ)

    Reviewers asked about the difference between abductive reasoning and link prediction, and whether naive methods could be applied. We clarified that abductive reasoning generates multi-hop logical hypotheses, while link prediction handles only simple edges, and search-based methods are computationally infeasible for large KGs. Our analyses show these naive approaches are insufficient, motivating specialized methods like CtrlHGen.

- **Others**(Reviewer MTp7 and Reviewer y9eJ)

    The reviewers raised questions regarding the applicability of our method across different KG datasets, its performance on large-scale graphs, and the inclusion of more insightful case studies. Through extensive experiments and detailed analysis, we have shown that our method is effective at any KG scale, including very large graphs. Additionally, the new case studies suggested by Reviewer y9eJ Q1 further highlight the practical potential and broader applicability of our approach.

---

### Meta-Review · Area_Chair_Bg9Z · 2026-01-03

**Summary:**

This paper introduces CtrlHGen, which aims to generate logically consistent hypotheses from a given premise. The authors introduced controllable hypothesis generation, which allows users to specify constraints on both the semantic content and the structural complexity of the desired hypotheses. They introduced a sub-logical decomposition strategy to augment the dataset, which helps the model learn complex structures. They designed a novel reward function that combines smoothed semantic rewards with a condition-adherence reward to ensure generated hypotheses are both accurate and follow the specified constraints.

The key concerns from the reviewers were resolved by the rebuttal. The left ones are not outstanding.

**Reviewer Concerns:**

The concerns (adding more baseline like Logic-Gen, adding advanced LLMs with thinking mode like GPT-OSS-5, etc.) from reviewer e8d8 were mostly resolved.

The concerns (requiring clear problem definition, concern of the generalizability of the proposed method to other KGs, requiring qualitative analysis of generated hypotheses from CtrlHGen) from reviewer MTp7 were mostly resolved.

Some concerns (complexity of the training pipeline and reproducibility of the results, comparing AbductiveKGR equipped with condition tokens with CtrlHGen) from reviewer y9eJ were resovled.  While other concerns (Scalability to large-scale KGs, baseline LLMs' failure may be partially due to this weak context-providing method) were partially resolved and are not outstanding.

The concerns from reviewer wmXZ were resolved.

**Reviewer Scores:**

Two reviewers gave 2 negative scores (both 4s) and two reviewers gave two positive scores (6 and 8). Reviewer e8d8 might raise their score as their concerns were mostly resolved.

---

### Decision · Program_Chairs · 2026-01-26

Accept (Poster)